# Fluctuations in chromatin state at regulatory loci occur spontaneously under relaxed selection and are associated with epigenetically inherited variation in *C. elegans* gene expression

Rachel Wilson [1,2,3], Maxime Le Bourgeois [3], Marcos Perez [3], Peter Sarkies [3]*

**1** MRC London Institute of Medical Sciences, London, United Kingdom, **2** Institute of Clinical Sciences, Imperial College London, London, United Kingdom, **3** Department of Biochemistry, University of Oxford, Oxford, United Kingdom

* peter.sarkies@bioch.ox.ac.uk

**Data Availability Statement:** All raw sequencing data has been uploaded to GEO and is accessible via the link https://www.ncbi.nlm.nih.gov/geo/

## Abstract

Some epigenetic information can be transmitted between generations without changes in the underlying DNA sequence. Changes in epigenetic regulators, termed epimutations, can occur spontaneously and be propagated in populations in a manner reminiscent of DNA mutations. Small RNA-based epimutations occur in *C. elegans* and persist for around 3–5 generations on average. Here, we explored whether chromatin states also undergo spontaneous change and whether this could be a potential alternative mechanism for transgenerational inheritance of gene expression changes. We compared the chromatin and gene expression profiles at matched time points from three independent lineages of *C. elegans* propagated at minimal population size. Spontaneous changes in chromatin occurred in around 1% of regulatory regions each generation. Some were heritable epimutations and were significantly enriched for heritable changes in expression of nearby protein-coding genes. Most chromatin-based epimutations were short-lived but a subset had longer duration. Genes subject to long-lived epimutations were enriched for multiple components of xenobiotic response pathways. This points to a possible role for epimutations in adaptation to environmental stressors.

## Author summary

Evolution is known to occur because of changes in DNA sequence which are inherited between generations. Recently, though, it has been discovered that information beyond the DNA sequence can be transmitted between generations. This information, known as epigenetic, can control how the DNA sequence is used. Epigenetic information that is transmitted between generations could drive evolutionary processes in populations, but this is yet to be tested. We used a simple nematode worm to investigate the contribution of different types of epigenetic information to evolution. We evolved populations of

query/acc.cgi?acc=GSE211846. The code has been uploaded to GitHub and is available via https://github.com/SarkiesLab/ChromatinEpimutations. Processed data used for the R code are available as .Rdata files and .csv files included in the supporting information. The processed data used for figures and tables in the manuscript are available as supporting information.

**Funding:** This work was funded by the Medical Research Council (to PS; Epigenetics and Evolution), the EPA Cephalosporin Trust (to PS) and the University of Oxford Department of Biochemistry (to PS). The salary of RW was funded by an MRC Chain-Florey Clinical PhD studentship at the London Institute of Medical Sciences. The Funders had no role in study design, data collection and analysis, decision to publish, or preparation of the manuscript.

**Competing interests:** The authors declare that we have no competing interests.

worms in the laboratory and investigated epigenetic differences that emerged in different lineages. Most epigenetic differences were very short-lived, so unlikely to be able to contribute to long-term evolutionary processes. However, we identified some changes that lasted much longer. Intriguingly, genes that control the worms' responses to external threats such as bacterial infections or noxious chemicals were most likely to undergo long-term epigenetic changes, despite the fact that the environment of the worms was stable and did not contain these stresses. We think that epigenetic processes might be able to create a fast-acting form of variation that could help in situations where organisms need to adapt to dangerous environments.

## Introduction

Epigenetic gene regulation is fundamental to the establishment and maintenance of cell identity during development. Whilst many epigenetic features are erased in germ cell development and the early stages of embryonic development, some transgenerational epigenetic inheritance phenomena have been rigorously established across a wide variety of experimental settings in several different species [1–5]. Several molecular mechanisms capable of transmitting epigenetic information between generations have been demonstrated, including short (18-36nt) non-coding RNA species, histone post-translational modifications, methylation of specific bases within DNA and prions [4,6]. Despite this burgeoning understanding of transgenerational epigenetic inheritance in laboratory settings we still understand little about its significance across evolution [7–9]. Of particular interest is whether transgenerational epigenetic inheritance can contribute to evolutionary processes in populations, including genetic drift and natural selection [5,10].

Heritable changes in epigenetic states may alter gene expression patterns but do not require underlying changes in DNA sequence [11]. When such changes arise in populations they can be referred to as epimutations [12,13]. Epimutations may be induced in response to an environmental stimulus [14–17], but have also been shown to arise spontaneously under controlled conditions [18,19]. Establishing the rate and stability of epimutations requires the use of approaches from evolutionary biology designed to map DNA sequence-based mutations, known as Mutation Accumulation (MA) experiments [20–22]. In the MA approach multiple lines derived from a common ancestor are propagated for many generations at reduced population size. Under these conditions drift dominates over selection so neutral, deleterious and beneficial mutations can be identified [20,23,24]. The MA framework was used to study epimutations in plants, focussing on changes in DNA methylation [25–27]. Spontaneous changes in DNA methylation arise more rapidly than DNA sequence changes, but have limited stability, lasting around 5–10 generations on average [18,28,29].

We recently used the nematode *C. elegans* as a model to study epimutation rate, spectrum and stability. Transgenerational epigenetic inheritance in *C. elegans* can be mediated by a class of small RNAs known as 22G-RNAs, which are made by RNA dependent RNA polymerase and map antisense to protein-coding genes [30–34]. We demonstrated that 22G-RNA-based epimutations arise spontaneously in *C. elegans* MA lines [35]. Epimutations were short-lived, lasting for two to three generations on average, although some epimutations lasted for at least 10 generations. Similar short-term variation was shown to affect metabolite levels, suggesting that short-term epigenetic inheritance could be influential in phenotypic variation [36]. This suggests that epimutations are unlikely to contribute to long-term divergence within populations but could contribute to short-term adaptive processes. However, there was no

enrichment of epimutations in 22G-RNAs to target particular categories of genes, thus the functional relevance of this process remains unclear [35].

In addition to 22G-RNAs, other forms of epigenetic regulation could contribute to epimutations in *C. elegans*. Chromatin is critical to the regulation of gene expression and is structurally dynamic in terms of its susceptibility to undergo remodelling in development and in response to external stimuli [37,38]. The preeminent features of chromatin organisation on *C. elegans* autosomes are large blocks of constitutively active genes marked by H3K36me3-containing nucleosomes and tissue-specific genes marked by H3K27me3 [37,39]. Additionally, transposable elements are marked by H3K9me2/3 nucleosomes [40]. Chromatin modifying enzymes contribute to transgenerational epigenetic inheritance phenomena either downstream or in parallel to small RNAs [41–46]. Here, we used the MA framework to investigate chromatin-based epimutations using ATAC-seq [47,48] to investigate large-scale changes in chromatin organisation. We demonstrate that heritable epimutations in chromatin structure occur, and are linked to changes in gene expression. Although most epimutations are very short-lived, a subset of epimutations is more durable. Intriguingly, long-lived epimutations are enriched for genes involved in defence response functions, suggesting that epimutations may have consequences for short-term adaptive responses for *C. elegans* in natural environments.

## Results

### 1. Spontaneous chromatin epimutations occur under relaxed selection

We reasoned that in addition to 22G-RNAs, chromatin-based epimutations could also be implicated in transmission of novel gene expression patterns through the germline (Fig 1A). Chromatin-based epimutations could therefore be a source of potential phenotype variation, which depending on stability and duration could be acted on by selective forces [5,10,49].

We propagated three independent Mutation Accumulation lines of the laboratory strain N2 *C. elegans* for 20 generations under identical environmental conditions. The population size was limited to two hermaphrodite animals to minimise selective forces [20] (Fig 1B). We collected embryos every two generations and profiled chromatin accessibility using ATAC-seq, small-RNAs using small RNA-seq and protein-coding genes using polyA-RNA-seq.

To identify genes with altered gene expression or antisense 22G-RNA levels we obtained normalized expression and 22G-RNA counts for each protein-coding gene and used a linear model to calculate deviations from the parental line. We then transformed the deviation into a Z-score for each gene for each generation across each lineage (S1–S6 Tables). To identify ATAC-seq changes we obtained accessibility information for regulatory elements on the basis of a previously published set [50]. We used a linear model to compare this to the parental line and transformed this into a Z-score as above (Fig 1C and S7–S9 Tables).

In order to identify putative epimutations from these data, we sought to identify a Z-score that maximised the probability of identifying true epimutations rather than false positives. According to the conventional definition of epigenetic inheritance [1], only inherited changes appearing in multiple successive generations are true epimutations. However, some epigenetic changes that appear in successive generations may have arisen by chance in the same locus over consecutive generations without transmission between generations, which we term false positive epimutations. We cannot distinguish these in any individual case. However, if true epimutations are common then we would expect the proportion of identical epimutations appearing in consecutive generations to be greater than expected given the rate at which epimutations occur overall. At a low threshold for defining an epimutation, there will be more false positives, but increasing the stringency will result in fewer true epimutations being detected. Across all Z-scores tested, the observation of chromatin state changes in two or more

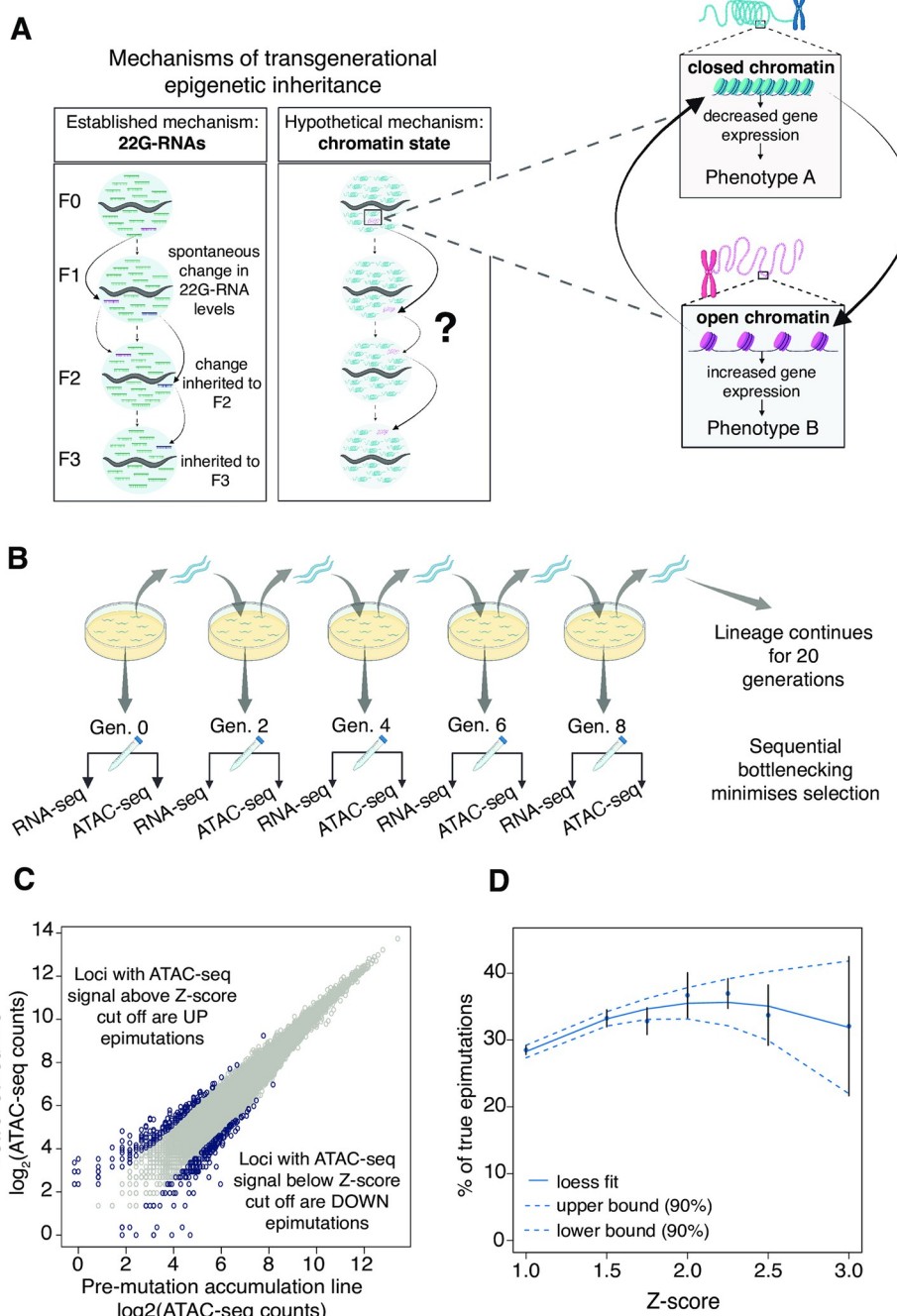

**Fig 1. Identification of spontaneous chromatin-based epimutations under relaxed selection. A.** Mechanisms of transgenerational epigenetic inheritance. 22G-RNAs have been established in previous work as drivers of epigenetic inheritance through the germ line. Spontaneous changes in chromatin state (epimutations) are an alternate mechanism. **B.** Experimental design. Three independent lineages of *C. elegans* nematodes were grown with bottlenecking for a total of 20 generations including a non-restricted pre-mutation-accumulation (PMA) generation (Gen. 0). Paired RNA-seq and ATAC-seq libraries were produced from the PMA generation and at alternate generations from generations 2–20. RNA was further processed to produce small RNA libraries. **C.** Identification of chromatin-based epimutations. Epimutated loci were defined as regulatory element loci with ATAC-seq counts (rings on plot) that were significantly greater (UP) or lower (DOWN) than that of the same locus in the PMA generation according to a Z-score cut off. UP epimutations correspond to increased opening of chromatin while DOWN epimutations reflect chromatin closing. **D.** Simulation to identify optimal Z-score cut-off. X-axis shows range of Z-score cut-offs tested. Y-axis shows difference between the percentage of epimutations inherited for two or more generations compared to the predicted percentage if epimutations occurred randomly but were never inherited.

consecutive generations was significantly greater in the real data than in the simulated data and was at a maximum Z-score of 2.25 (Fig 1D). We set the threshold for identification of epimutations at 2.25 for subsequent analysis. Therefore, increased ATAC-seq signals (Z > 2.25) were 'UP' chromatin-based epimutations, indicating more open chromatin states. Decreased ATAC-seq signals (Z < -2.25) were 'DOWN' chromatin-based epimutations, indicating more closed chromatin states. We took the same approach to identify small RNA-based epimutations, for which 'UP' indicated increased small RNA levels while 'DOWN' indicated decreased small RNA levels.

## 2. Chromatin accessibility and 22G-RNA diversify at similar rates during evolution of *C. elegans* in the laboratory

We identified chromatin-based epimutations in three independent lineages propagated under minimal selection and compared their properties to epimutations in small non-coding RNAs and gene expression (S10–S18 Tables). We investigated microRNAs, piwi RNAs and 22G-RNAs separately. Heritable epimutations in miRNA expression levels were very rare, although interestingly slightly above what would be expected by chance. Heritable epimutations in piRNA expression levels were more common but affected fewer loci than changes in 22G-RNAs mapping to protein-coding genes (S1 Fig). On the basis of this and the known role of 22G-RNAs in transgenerational epigenetic inheritance in *C. elegans* [35], we focussed on 22G-RNA-based epimutations. On average, about 1.6% of protein coding genes exhibited changes in expression level per generation (Fig 2A). The median survival was 5, 5 and 3 generations for gene expression changes, 22G-RNA-based epimutations and chromatin-based epimutations respectively (Fig 2B). All three worm lineages displayed similar rates of epimutations (S2 Fig). This is orders of magnitude higher than the genome-wide rate of spontaneous nucleotide sequence change or insertion-deletion events in MA lines of *C. elegans*. [51–54].

## 3. Epimutations occur both with and without simultaneous changes in gene expression

We next asked whether changes in gene expression were accompanied by simultaneous changes in chromatin accessibility or 22G-RNA levels. We reasoned that simultaneous epimutations might be mechanistically linked to expression changes as outlined in the diagram (Fig 3A). To test this, we conducted a stepwise analysis of the association between chromatin-based epimutations and expression changes. We first looked at a background of all genes with annotated regulatory elements regardless of whether they had expression changes or not. The proportion of the genes which had expression changes was compared to the proportion of the genes which had chromatin-based epimutations. The overlap, in which genes had both expression changes and epimutations, was calculated (S56 Table). At this level, we found that chromatin-based epimutations rarely accompanied gene expression changes (Tests 1 & 2 in Table 1).

We then looked more closely within the subset of genes which had expression changes. In this background we found a strong association between inherited expression changes and simultaneous chromatin-based epimutations (Fig 3B and Tests 3 & 4 in Table 1).

Using the equivalent stepwise approach for 22G-RNAs (S57 Table), we found that changes in 22G-RNAs mapping antisense to protein-coding genes overlapped more strongly with changes in gene expression in a background of all genes (Test 1 in Table 2), and as with chromatin, genes showing heritable changes in gene expression were significantly enriched for simultaneous 22G-RNA-based epimutations (Fig 3C, Tests 3 & 4, Table 2).

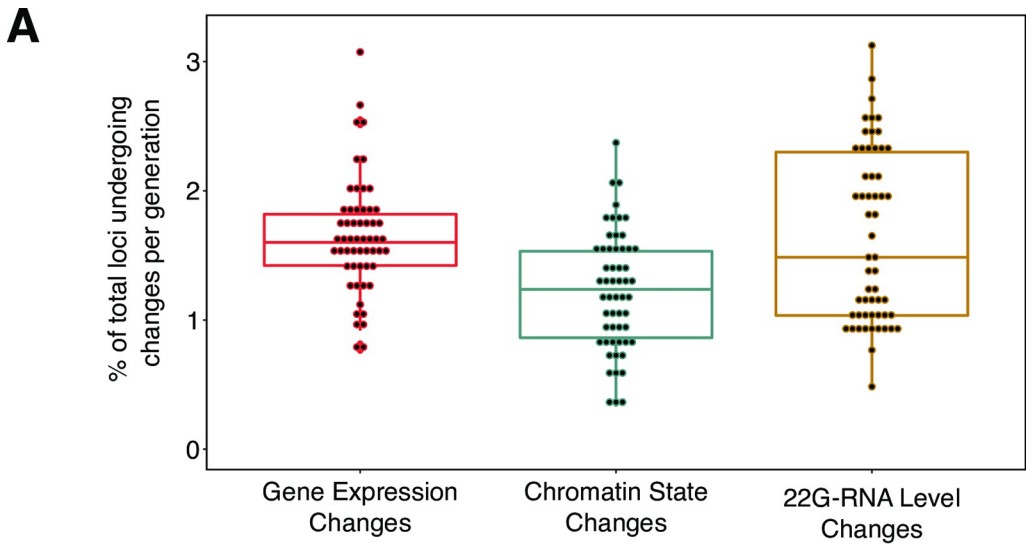

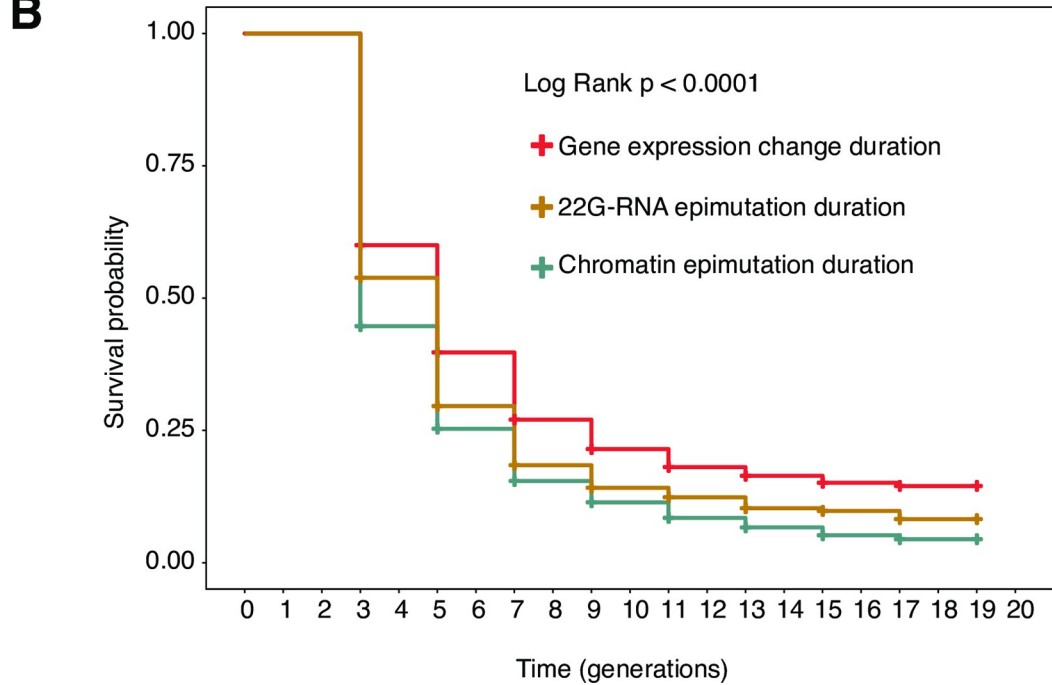

**Fig 2. Rate of onset and survival of gene expression changes, chromatin and 22G-RNA-based epimutations. A.** Boxplots indicate percentage of loci per generation subject to gene expression changes (red), chromatin state changes (blue) or 22G-RNA level changes (gold). Box shows the interquartile range with horizontal line at the median; the whiskers extend to the furthest point no more than 1.5 times the interquartile range **B.** Survival plot shows step changes in survival of gene expression changes (red), 22G-RNA-based epimutations (gold) and chromatin-based epimutations (blue) persisting over a time frame of 20 generations (generational time points on X-axis). Y-axis showing survival probability. p-value calculated with log rank test.

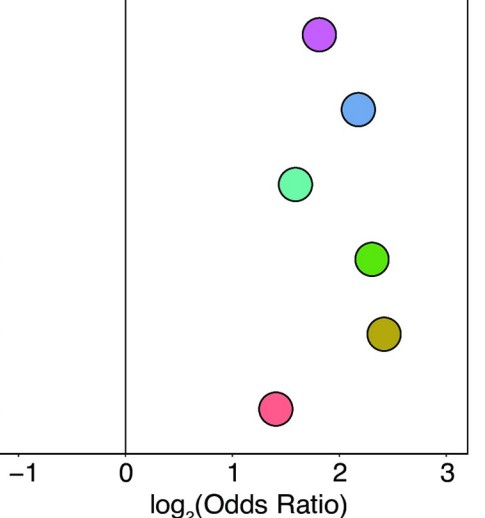

**Fig 3. Genes with inherited expression changes are enriched for simultaneous chromatin-based epimutations. A.** Chromatin-based epimutations which alter accessibility of regulatory elements to transcription factors (TF) and simultaneous gene expression changes may be inherited in parallel. **B.** Stacked bar-plot showing proportions of inherited (left bar) and non-inherited (right bar) gene expression changes accompanied by simultaneous chromatin-based epimutations which may themselves be inherited (pink) or non-inherited (blue). **C.** Stacked bar-plot showing proportions of inherited (left bar) and non-inherited (right bar) gene expression changes accompanied by simultaneous 22G-RNA-based epimutations which may themselves be inherited (blue) or non-inherited (beige). **D.** Examination of concordance (simultaneous and direction matched) of gene expression changes and chromatin-based epimutations. Y-axis shows associations tested. X-axis shows log$_2$(Odds) of gene expression changes within each association being inherited. Odds ratios and p-values calculated with Fisher's Exact Test. **E.** Examination of concordance (simultaneous and direction matched) of gene expression changes and changes in 22G-RNAs. Y-axis shows associations tested. X-axis shows log$_2$(Odds) of gene expression changes within each association being inherited. Odds ratios and p-values calculated with Fisher's Exact Test. p-value cut off for significance is 0.1.

**Table 1. Stepwise analysis of association of gene expression changes and chromatin-based epimutations.** Cells in blue indicate significantly reduced odds of associa-tion, cells in grey indicate no significant association, cells in yellow indicate significantly increased odds of association. Odds ratios and p-values calculated with Fisher's Exact Test.

| Test | Association tested | Background | Odds Ratio | p-value |
|---|---|---|---|---|
| 1 | Gene expression change &Chromatin epimutation | All genes | 0.59 | $3.31 \times 10^{-40}$ |
| 2 | Inherited gene expression change & Chromatin epimutation | All genes | 1.08 | $3.4 \times 10^{-01}$ |
| 3 | Inherited gene expression change & Simultaneous chromatin epimutation | All genes with RNA-seq changes | 3.57 | $3.34 \times 10^{-28}$ |
| 4 | Inherited gene expression change & Simultaneous chromatin epimutation | All genes with RNA-seq changes and chromatin state changes | 7.65 | $1.89 \times 10^{-40}$ |

Therefore, although chromatin-based epimutations may occur without concurrent expres-sion changes, on occasions where heritable expression changes are observed, simultaneous chromatin-based epimutations tend to accompany these. Chromatin-based epimutations may produce a favourable environment for changes in gene expression to be inherited but addi-tional factors such as 22G-RNAs may be required to 'license' these.

We examined whether chromatin-based epimutations occurring in the same generation (simultaneous) and in the same direction (UP/UP or DOWN/DOWN) as gene expression changes i.e. 'concordant' had any effect on the odds that expression changes would be inher-ited, compared to simultaneous but 'discordant' (UP/DOWN, DOWN/UP) changes. Surpris-ingly, we found that genes with both concordant and discordant chromatin-based epimutations were similarly enriched for inherited expression changes (Fig 3D) which was in keeping with previous findings made from 22G-RNAs described above.

In the case of genes with simultaneous 22G-RNA-based epimutations and gene expression changes, we found that both concordant (UP/UP or DOWN/DOWN) and discordant (UP/DOWN, DOWN/UP) 22G-RNA-based epimutations were similarly enriched for inherited expression changes (Fig 3E). The enrichment for discordant 22G-RNA mediated epimutations was consistent with our previous observations [35] and the function of 22G-RNAs in gene silencing [55,56]

## 4. Chromatin environment affects the duration of epimutations

Previous work has shown that inherited epigenetic effects are typically short-lived. However, a small proportion of epimutations have been shown to persist over 10 or more generations [35]. In agreement with these results, we found that the duration of the majority of epimuta-tions in chromatin, 22G-RNAs and gene expression was around 3–5 generations (Fig 2B). However, some epimutations lasted considerably longer, in some cases over 10 generations. We used K-means clustering with two clusters to identify a subset of loci that were subject to long-lived epimutations (Methods; S3A–S3C Fig) and showed that these strongly overlapped with loci derived by fitting the distribution to two normal distributions (Methods; S3D–S3F Fig). We compared these to heritable changes that were not in the long-lived set, which

**Table 2. Stepwise analysis of association of gene expression changes and 22G-RNA-based epimutations.** Cells in grey indicate no significant association, cells in yellow indicate significantly increased odds of association. Odds ratios and p-values calculated with Fisher's Exact Test.

| Test | Association tested | Background | Odds Ratio | p-value |
|---|---|---|---|---|
| 1 | Gene expression change & 22G-RNA epimutation | All genes | 2.06 | $4.42 \times 10^{-28}$ |
| 2 | Inherited gene expression change & 22G-RNA epimutation | All genes | 1.1 | $6.04 \times 10^{-01}$ |
| 3 | Inherited gene expression change & Simultaneous 22G-RNA epimutation | All genes with RNA-seq changes | 2.29 | $6.19 \times 10^{-07}$ |
| 4 | Inherited gene expression change & Simultaneous 22G-RNA epimutation | All genes with RNA-seq changes and 22G-RNA changes | 3.55 | $3.56 \times 10^{-09}$ |

comprised loci with "short-lived" changes, and other loci that showed changes which were non-inherited. We examined the chromatin environments of these sets of genes.

Genes with expression changes and 22G-RNA-based epimutations were enriched in Regulated domains (marked by H3K27me3) and depleted from Active domains marked by H3K36me3 [39] (Fig 4A and 4C). The X chromosome, which is subject to H4K20 methylation as a result of dosage compensation and does not show high levels of either H3K27me3 or H3K36me3 [57–59], was also enriched for both non-inherited and heritable short-lived 22G-RNA-based epimutations (Fig 4C).

Contrastingly, chromatin-based epimutations, regardless of duration, were enriched in Active domains (Fig 4B). Interestingly, we noticed that piRNA clusters, which occupy two broad regions on chromosome IV [60–62] were enriched for non-inherited chromatin-based epimutations (Fig 4B). This was surprising because piRNAs are predominantly produced from H3K27me3-marked regions [63]. Indeed, the proportion of chromatin epimutations in the piRNA clusters compared to the wider genome was greater than that of 22G-RNA-based epimutations and gene expression changes (S4 Fig). Stratification of epimutations by whether they were UP or DOWN did not affect the overall trends, suggesting that both directions contribute to the enrichments observed for all types of epimutation analysed (S5 Fig).

In previous work we investigated stability of 22G-RNA-based epimutations targeting genes associated with different small RNA pathways [35]. We showed that epimutations targeting the CSR-1 pathway, which is associated with gene activation [64], were unstable and short-lived, while those targeting the WAGO-1 pathway, which drives gene silencing [55], were highly stable, at times persisting beyond 10 generations. Having identified 22G-RNA-based epimutations with differing durations in this work, we then checked for differential associations with various small RNA pathways. We found the same enrichment pattern as in our previous work. Targets of non-inherited 22G-RNA-based epimutations strongly overlapped with CSR-1 pathway targets while targets of long-lived 22G-RNA-based epimutations strongly overlapped with WAGO-1 pathway targets (S6 Fig). Epimutations targeting both activating and silencing small RNA pathways is a possible explanation for why gene expression changes were similarly enriched for both concordant and discordant 22G-RNA-based epimutations (Fig 3E).

## 5. Long-lived epigenetic changes arise in genes functioning in adaptive responses

We next investigated whether epimutated genes with different durations of inheritance were enriched for particular biological functions. Genes with significantly long-lived expression changes were enriched for ontology terms relating to defence responses against bacteria and radiation (Fig 5A). This hints at a role in potentially adaptive processes [65] Compared to non-inherited and short-lived categories, long-lived chromatin-based epimutations at regulatory elements were strongly enriched across an array of pathways (Fig 5B), including regulation of defence response to bacterium. This finding is surprising given the previous observation (Fig 4A and 4B) that gene expression changes and chromatin epimutations are predominantly in distinct chromatin domains. Additionally, we were interested to see enrichment in germ-line stabilising SET domain pathways [66], and other pathways involving translation, protein localization and development. This supports the notion that chromatin-based epimutations establish conducive environments for heritable expression changes to occur which may have wide ranging phenotype effects. Genes with long-lived 22G-RNA mediated epimutations, however, did not show enrichment for terms relating to bacterial defence responses (Fig 5C).

We performed an ontology enrichment analysis of the set of genes that had inherited expression changes and simultaneous chromatin epimutations compared to genes which had

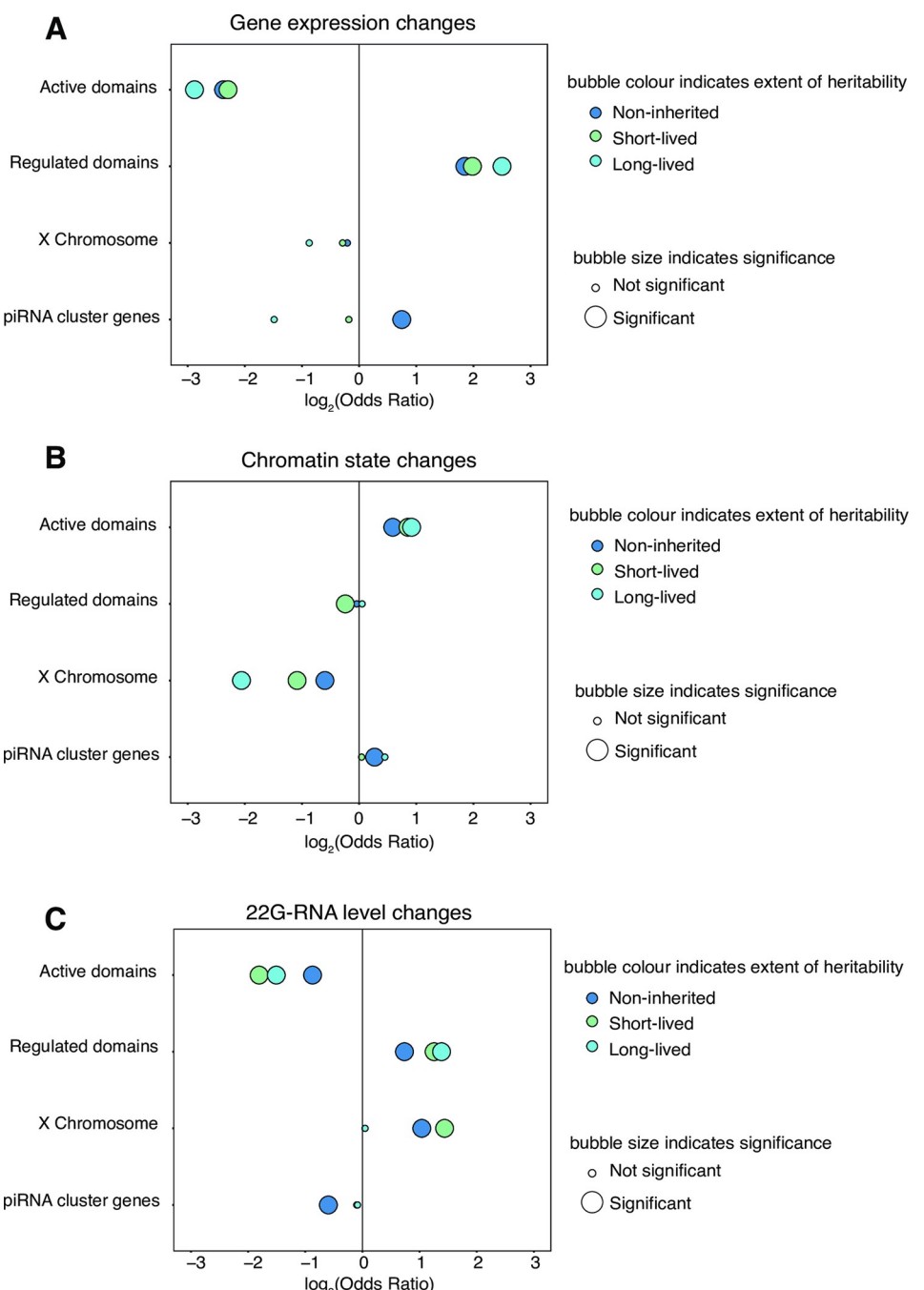

**Fig 4. Enrichment of gene expression changes and epimutations in different constitutive chromatin domains. A.** Bubble plot showing distribution of non-inherited (dark blue), short-lived (light green) and long-lived (light blue) gene expression changes in distinct chromatin domains. **B.** Bubble plot showing distribution of non-inherited, short-lived and long-lived chromatin-based epimutations in distinct chromatin domains. **C.** Bubble plot showing distribution of non-inherited, short-lived and long-lived 22G-RNA-based epimutations in distinct chromatin domains. For all plots, Y-axis shows constitutive chromatin domains investigated. X-axis shows $\log_2$(Odds) of enrichment. Odds ratios and p-values calculated with Fisher's Exact Test with Bonferroni correction. p-value cut off for significance is 0.1.

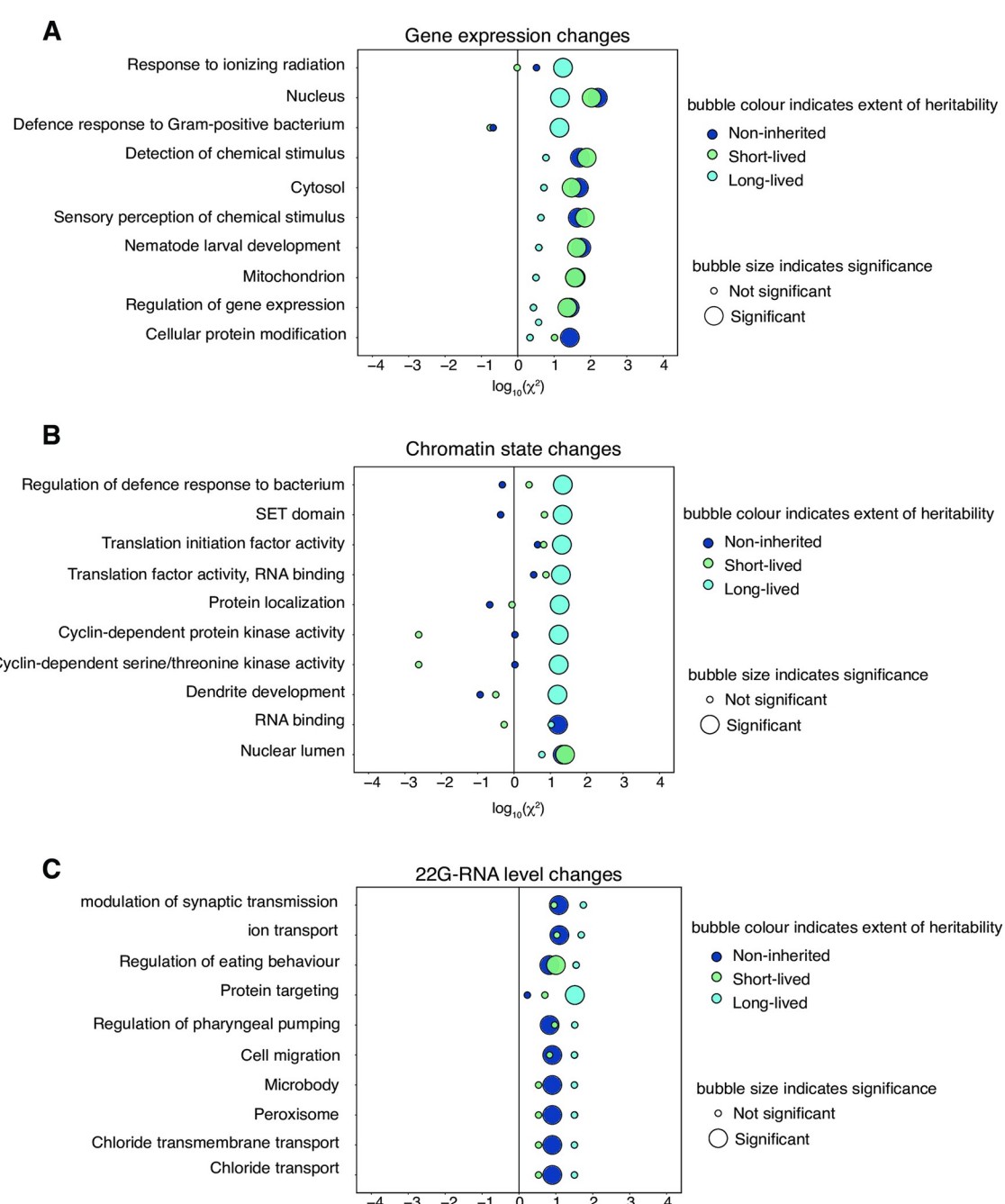

**Fig 5. Genes with long-lived epigenetic changes are enriched for nematode defence functions. A.** Bubble plot showing ontology term enrichment of genes with non-inherited (dark blue), short-lived (light green) and long-lived (light blue) expression changes compared to genes without expression changes. Enrichment calculated with $\chi$-squared test, top 10 results shown. X-axis shows $\log_{10}(\chi)$ for enrichment. Y-axis shows ontology terms. **B.** Bubble plot showing ontology term enrichment of regulatory element loci with long-lived, short-lived and non-inherited chromatin-based epimutations compared to genes without chromatin-based epimutations. Enrichment calculated with $\chi$-squared test, top 10 results shown. X-axis shows $\log_{10}(\chi)$ for enrichment. Y-axis shows ontology terms. **C.** Bubble plot showing ontology term enrichment of genes with long-lived, short-lived and non-inherited 22G-RNA-based epimutations compared to genes without 22G-RNA-based epimutations. Enrichment calculated using Fisher's Exact Test with Bonferroni correction, top 10 results shown. X-axis shows $\log_{10}(Odds)$ of enrichment. Y-axis shows ontology terms. For all plots, p-value cut off for significance is 0.1.

non-inherited gene expression changes and simultaneous chromatin epimutations. Interestingly, this gene set was enriched for genes implicated in bacterial defence functions (S7 Fig).

### 6. Four P-glycoprotein genes are in a potential operon and have expression changes which may be explained by simultaneous 22G-RNA-based epimutations

Amongst genes with long-lived changes in expression, we identified 4 P-glycoprotein (*pgp*) genes. This was particularly interesting as PGP proteins have key functions in defence against xenobiotics, including a key role in drug resistance both in humans and nematodes [67–69]. *Pgp* genes 5, 6, 7 and 8 had long-lived epigenetic changes in all three lineages. The expression patterns for the 4 *pgp* genes were strongly similar within each lineage (Fig 6A and 6B). The fact that *pgp* expression was most often increased relative to the parental lineage could be due to chance, but it could also suggest that this might reflect an unstable epimutation that occurred in the population prior to the onset of the experiment. The organisation of *pgp* genes 5, 6, 7 & 8 in a 4 gene cluster has been described previously [70] but this cluster has not been annotated as an operon [71].

Although not matching the threshold for epimutations, we observed that chromatin activity at the three regulatory elements for *pgp*-6 appeared to broadly track with some of the changes in gene expression for *pgp* genes 5, 6, 7, and 8 (Fig 6A). We wondered whether the genome-wide threshold for qualifying a chromatin state change as an epimutation under-estimated the extent of changes in the setting of the X chromosome due to its unique chromatin environment [59]. To address this, we derived Z-scores for X chromosome gene expression and chromatin state change loci using the same method as before but restricting the distribution of data points to those only derived from the X chromosome in order to produce X chromosome-specific Z-scores. However, the effect of adjusting for the specific context of the X chromosome was minimal (S8 Fig). Interestingly, when we examined the behaviour of 22G-RNA-based epimutations targeting genes within the putative *pgp* operon, a steep decline in 22G-RNAs occurred at precisely at the same time as the initial steep increase in gene expression (Fig 6B). In subsequent generations, the fluctuations in chromatin state changes appeared to correlate with gene expression changes (Fig 6A). However, it is important to note that neither the 22G-RNAs nor changes in chromatin correlate strongly with gene expression changes, suggesting that additional regulatory factors may be involved in regulating the heritable changes in gene expression that we observed at these loci.

## Discussion

Here we have demonstrated that epimutations arise at the level of chromatin during long-term propagation of *C. elegans* under conditions of relaxed selection. These epimutations are heritable, and, in some cases, correspond to changes in gene expression. This expands our previous work demonstrating that 22G-RNA levels are subject to spontaneous short-lived epimutations in *C. elegans* [35] It also fits with recent results showing that short-term heritable variation in metabolite levels occurs in C. elegans lines propagated with limited population size [36]. Additionally, we show that epimutated genes display biased localisation across chromatin domains, and are enriched for genes with particular functional characteristics. Below, we discuss the implications of our findings for our understanding of the mechanism of epimutation formation and the potential consequences for evolutionary processes occurring within populations.

Our data shows clearly that spontaneous changes in gene expression arise during long-term propagation of *C. elegans* lineages. These changes are mostly short-lived, although occasionally long-lived epimutations appearing in more than 10 consecutive generations were observed.

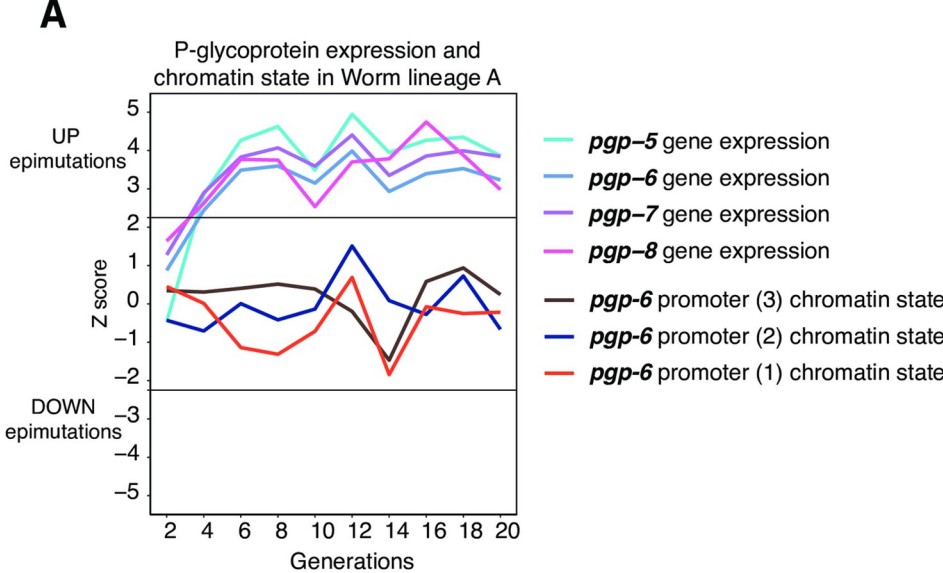

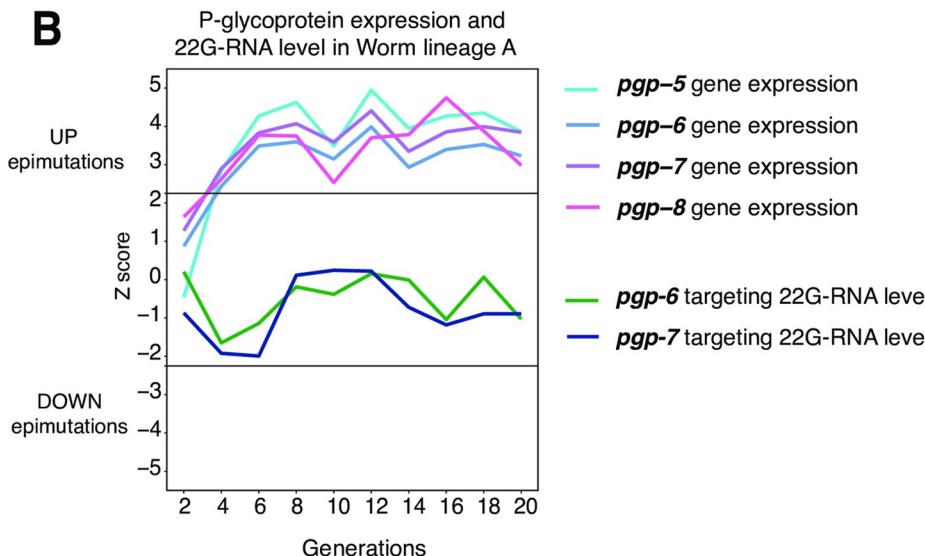

**Fig 6. A four gene cluster of P-glycoprotein genes has long-lived RNA-seq changes and may be co-regulated. A.** Chromatin states of three *pgp-6* promoters and *pgp-5*, *pgp-6*, *pgp-7*, *pgp-8* gene expression trends over 20 generations in Lineage A. Generational time points on X-axis. Z-score for epimutation status shown on Y-axis with 0 equivalent to PMA state. Horizontal thresholds indicate Z-score cut-offs; > 2.25 = UP epimutation, < - 2.25 = DOWN epimutation. **B.** Levels of 22G-RNAs targeting *pgp-6* and *pgp-7* over 20 generations in Lineage A. Generational time points on X-axis. Z-score for epimutation status shown on Y-axis with 0 equivalent to PMA state. Horizontal thresholds indicate Z-score cut-offs; > 2.25 = UP epimutation, < - 2.25 = DOWN epimutation.

Changes in 22G-RNAs may account for around 26% of the heritable changes. Although significantly greater than expected by chance, changes in chromatin accessibility explain only around 12% of the remaining heritable changes. Interestingly, we also observe many changes in chromatin accessibility at regulatory elements that do not correspond to changes in the expression of nearby genes. Thus, changes in chromatin accessibility are to some extent

decoupled from gene expression changes [72,73]. We note that in some cases this might reflect the fact that subtle changes, not sufficient to qualify as an epimutation, may be responsible-indeed we observed potential evidence for such changes at clustered *pgp* gene loci (Fig 6). Nevertheless, it seems likely that considerable changes in chromatin accessibility, which are heritable trans-generationally, can occur without changes in gene expression. Potentially the redundant structure of gene expression regulation in metazoans, whereby several regulatory elements combine to regulate each gene [74,75] could mean that spontaneous changes in one regulatory element are rarely sufficient for gene expression changes to follow. The fact that we find both concordant and discordant changes in chromatin and gene expression further emphasises that the coupling between chromatin accessibility and gene expression can be complex [73].

As we used ATAC-seq to map changes in chromatin accessibility, we are unable to determine the precise molecular events that occur during initiation or propagation of chromatin-based epimutations. However, the ability of the changes we detect to be sustained over multiple generations without obvious changes in transcription are consistent with the proposal that histone modifications can be transmitted by the ability of histone modifying enzymes to recognise the modifications that they themselves introduce [76]. Further investigation using ChIP-seq to identify changes in candidate histone modifications will be required to decipher the histone modifications involved and suggest possible modes of propagation.

An important remaining question is what underpins the remainder of the heritable changes in gene expression that cannot be explained by changes in chromatin or 22G-RNAs. It is possible that chromatin changes affecting specific histone modifications might not manifest as changes in ATAC-seq signal despite being linked to local changes in gene expression. Although *C. elegans* does not have cytosine DNA methylation [77] and no mechanism to transmit adenine DNA methylation is known [78], other sources of epigenetic information might be important, including prions and modifications to RNA molecules. Another interesting possibility is that RNA itself could act as an epigenetic signal. Random accumulation of additional mRNA molecules due to aberrant transcription during germ cell development might be sufficient to trigger a long-term propagation of a gene expression change, due to the ability of maternally deposited RNAs to "licence" subsequent expression [79].

An important controversy in the field of epigenetic inheritance is whether epigenetic variation produces phenotype variation that could be subject to selection [5,10]. Here we have shown that long-lived epigenetic changes arise in genes with key roles in xenobiotic detoxification. Long-lived chromatin-based epimutations also occurred in regulatory elements acting on xenobiotic response genes (Table 3). These genes showed different changes in different lineages and epimutations in these genes occurred at different time points within individual lineages. This suggests that these epimutations occur spontaneously even under stable environmental conditions, although it remains possible that small, environmental fluctuations outside of our control may be responsible for some of these alterations.

Epimutations arising in these genes in stable environments could generate diversity in xenobiotic response profiles within a population. Xenobiotic defence gene families (Table 3) show a marked propensity for significant up and down fluctuations in expression over 20 generations while no such fluctuations are seen in a set of known housekeeping genes [81] (S9 Fig). Such fluctuations are reminiscent of a 'bet hedging strategy' whereby offspring have mixed fitness under different conditions. Stochastic and long-lived epimutations might enable epigenetic 'anticipation' of many possible environments [82,83]. The short-lived nature of epimutations may enable populations to partially adapt to environmental stress before adaptive genetic mutations have had time to emerge [5,10]. Further work is required to determine if innate epimutations could be subject to selection, and subsequently if epigenetically driven phenotypes become genomically encoded, through the effect of 'genetic takeover' [5,84].

**Table 3. Four gene families with critical roles in xenobiotic defence have long-lived epimutations.** Xenobiotic defence genes are represented in the subset of genes with long-lived expression changes and long-lived chromatin-based epimutations but only one such gene had long-lived 22G-RNA-based epimutations.

| Gene family | Function in xenobiotic defence | With long-lived expression changes | With long-lived chromatin epimutations | With long-lived 22G-RNA epimutations |
|---|---|---|---|---|
| *nhr* | Nuclear Hormone Receptors are transcription factors which regulate xenobiotic response [68,69]. | *nhr-165* <br> *nhr-81* | *nhr-41* <br> *nhr-19* <br> *nhr-203* <br> *nhr-170* | |
| *cyp* | Cytochrome P450 enzymes are active in phase 1 of xenobiotic detoxification [68]. | *cyp-14A5* <br> *cyp-35A1* <br> *cyp-35A4* <br> *cyp-35B2* | *cyp-29A3* <br> *cyp-33C9* | |
| *ugt* | UDP-glucuronosyltransferases mediate conjugation of drugs and pollutants, to glucuronic acid for removal. [65]. | *ugt-26* <br> *ugt-8* | *ugt-64* | *ugt-52* |
| *mrp/pgp* | Multidrug resistance protein and P-glycoprotein are subsets of the ABC transporter family which export inactivated xenobiotics [80]. | *pgp-5* <br> *pgp-6* <br> *pgp-7* <br> *pgp-8* <br> *pgp-11* | *mrp-7* <br> *mrp-8* | |

Under conditions of xenobiotic stress, the propagation of pre-existing epimutations which are now advantageous in this setting is enhanced due to selection. On removal of the stressor, epimutations no longer provide a survival advantage, or indeed may be detrimental. Selection then ceases to drive propagation of the epimutation (Fig 7). Alternatively, the mechanism of epimutation inheritance itself may become more robust under stress, for example due to enhanced 'self-templating' of the epimutation [76], leading to population expansion of an epimutation which may or may not have adaptive benefit. Enhanced heritability returns to baseline when the stressor is removed. Either or both of these mechanisms might enable epimutations to contribute to adaptation to xenobiotic stress. It will be intriguing to test whether this occurs in nematode populations exposed to stress conditions.

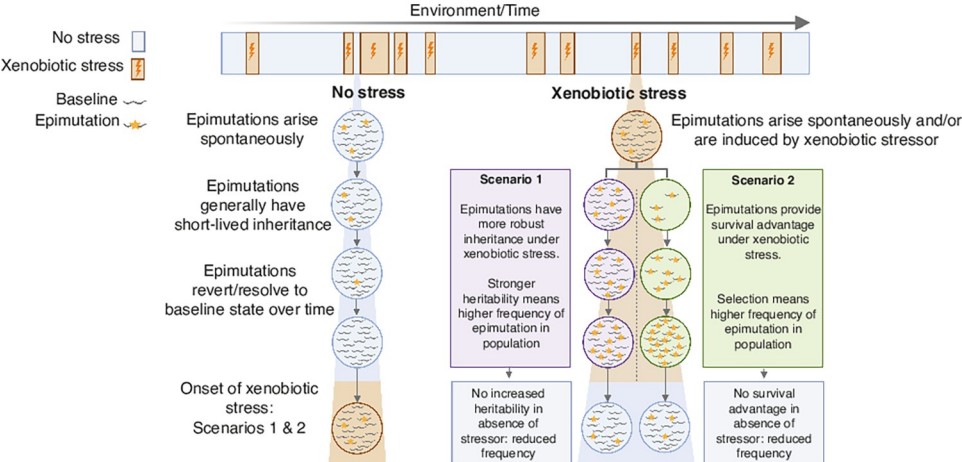

**Fig 7. Two potential mechanisms for propagation of spontaneous epimutations under xenobiotic stress.** We propose two scenarios whereby epimutations may be propagated with short term reversibility under conditions of xenobiotic stress and the potential for adaptive consequences.

## Methods

### Construction of mutation accumulation lines

*Caenorhabditis elegans* N2 worms were grown at 20°C in a temperature-controlled incubator with time out of the incubator kept to a minimum. Worms were grown on Nematode Growth Medium (NGM) plates with an OP50 *E. coli* lawn. The use of Mutation Accumulation (MA) lines has been described previously [52]. MA lines are produced by restricting the effective population size by selecting the minimal number of individuals required to found each successive generation. Trial studies confirmed that in order to have sufficient embryo material for ATAC-seq, two worms were required to found each new generation. Each of three independent lineages (A, B & C) was therefore founded by picking two N2 L4 hermaphrodite worms from an expanded population. On day 4, two L4 hermaphrodite worms were picked from the founder plate (pre-mutation accumulation, or generation zero) to a new plate to produce the subsequent generation. On day 5, embryos were collected from the original plate. Lineages were propagated in this way for 20 generations. Embryos were collected at generation zero and then at alternate even numbered generations; 0, 2, 4, 6, 8, 10, 12, 14, 16, 18, 20. For embryo collection, worms and embryos were washed off plates into 15 ml falcon tubes with 0.1% Triton X. Plates were washed repeatedly to maximise collection of worms and embryos. The falcon tubes were centrifuged at 2000 RPM for 1 minute and the supernatant removed. Remaining embryos on the washed plates were loosened from agar by spraying 0.1% Triton X with the pipette angled against the plate. The volume of liquid containing additional washed off embryos was added to the worm pellet. Three further washes of the pellet were performed until the supernatant was clear to ensure that OP50 bacteria had been removed. Bleaching whole worms destroys the worm, leaving embryos intact. This meant that only embryonic material was retained for epigenetic and RNA-seq analysis. Worms were bleached with hypochlorite treatment in which 5.5 ml of bleach was added to each sample. Samples were vortexed continuously for 5 minutes with intermittent checking to ensure worm carcasses were dissolving and to avoid over exposure of embryos to bleach. Bleach was washed out rapidly once worm carcasses had disappeared with addition of M9 to dilute bleach, centrifugation to pellet undissolved embryo material, complete removal of supernatant and repeating this procedure for a total of three washes. Following the 3 post bleaching washes, the supernatant was removed from the embryo pellet. The pellet was resuspended in 1 ml 0.1% Triton X and then split into 1.5 ml microcentrifuge tubes follows: 25% of each sample was reserved for RNA library preparation and 100 ul TRIzol was added to inhibit RNase and maintain RNA integrity. 75% of each sample was reserved for ATAC-seq and frozen with no additive. Microcentrifuge tube were then submerged in liquid nitrogen before being stored at– 80°C.

### Extraction of material for assessment of epimutations

**RNA extraction.** Chloroform Isopropanol extraction of RNA was performed following standard protocols which had been optimised for our lab [35,85] as follows; embryos were collected in TRIzol as described above. Embryos were lysed through 5 freeze-cracking cycles in which embryos were frozen in liquid nitrogen and then thawed in a 37°C water bath. Tubes were vortexed for 5 minutes with 30 second pauses at 30 second intervals, and then incubated at room temperature for 5 minutes. This causes disruption of RNA-protein complexes. 200 μl chloroform per ml of TRIzol were added to samples followed by vigorous shaking. Samples were then incubated for two to three minutes at room temperature followed by centrifugation at a maximum speed at 4°C for 10 minutes. The top aqueous layer was aspirated and transferred to a new tube into which 1 μl glycogen and an equal volume of isopropanol was added.

RNA was precipitated overnight at -20˚C. After overnight precipitation, the samples were centrifuged for 1 hour at 4˚C. The supernatant was removed and 500 μl of 75% ETOH added to the pellet. The samples were centrifuged at max speed for 10 minutes at 4˚C. All ETOH was removed and the pellet allowed to air dry. The RNA pellet was resuspended in 10 μl H20. RNA sample concentration and quality was quantified on a 2200 TapeStation instrument using Agilent RNA screen tapes. An RNA Intengrity (RIN) score was derived. Samples were additionally quantified using Nanodrop.

**Assay for Tranposase Accessible Chromatin (ATAC).**    The protocol was adapted from Daugherty [48] after the original method for ATAC-seq [47]. Nuclei were extracted from embryo samples through cycles of freeze-cracking as follows. 200 μl of Nuclear Preparation Buffer was added to each microcentrifuge tube containing frozen embryos. The sample were submerged in liquid nitrogen for 90 seconds and then transferred to a 37˚C water bath for 90 seconds. This was repeated three times. Embryo material was then transferred to a Wheaton glass tissue homogenizer in ice and was ground (3.5 grinds). The vessels containing the embryo material were covered and centrifuged for two minutes at 200 RCF at 4˚C. The supernatant containing nuclei was transferred to a microcentrifuge tube on ice. The remaining embryo material was ground again, spun down and the supernatant again transferred to the collection tube. This cycle was repeated 4 times. The collection tubes were spun at 1000 RCF at 4˚C for 10 minutes. The supernatant was discarded, leaving the nuclei pellet behind. To each sample nuclei pellet, Tagmentation enzyme and Tagmentation buffer were added (Illumina Tagment DNA enzyme and buffer). The samples were incubated for 30 minutes at 37˚C on a Thermoshaker set to 580 RPM. Samples then underwent DNA clean up with a Qiagen MiniElute Reaction Cleanup Kit. The resulting DNA was eluted in 10 μl of EB buffer. PCR adapters from the Illumina Nextera DNA prep kit (Illumina) were selected and added to each sample along with PCR master mix. PCR was run with the following cycle parameters: 72˚C for 5 minutes, 98˚C for 30 seconds, 14 cycles (98˚C for 10 seconds, 63˚C for 30 seconds, 72˚C for 1 minute). 4˚C hold. Size selection isolates the desired DNA fragments and was achieved with magnetic AMP X beads (Beckman Coulter). First, excessively large DNA fragments were removed as follows; 25 μl AMP X bead solution was added to each sample. Samples were incubated at room temperature for 10 minutes. The PCR tubes containing the bead and sample mix were then put onto a magnetic rack for 5 minutes until the sample appeared clear. The samples were transferred without disturbing the magnetic beads to a new set of PCR tubes. To remove excessively small DNA fragments, the same procedure was followed, but the volume of AMP X bead solution added was adjusted to 65 μl. After incubating on the magnetic rack, the clear liquid was carefully removed and discarded. At this point, the beads had the desired library bound. The bead pellet was washed twice with 80% EtOH and then allowed to air dry at room temperature for two minutes. Excess EtOH droplets were removed with a pipette tip from inside each PCR tube. 22 μl of nuclease free water was added to each sample and the beads were washed down and dispersed into the liquid. The PCR tubes were returned to the magnetic rack, the DNA having eluted into the nuclease free water. The liquid was removed without disturbing the magnetic beads to a final collection tube which could then be frozen at -80˚C. Validation of ATAC libraries was performed by quantification on the 2200 TapeStation using Agilent D1000 Screen tapes. ATAC libraries were also quantified on Qubit using the Qubit dsDNA HS Assay kit from Invitrogen.

**Small RNA extraction.**    The TruSeq Small RNA Prep Kit (Illumina) was used to prepare small RNA libraries from RNA pyrophosphohydrolase (Rpph) treated RNA. Rpph increases efficiency for 22G-RNA sequencing by converting 22G 5-prime triphosphate to monophosphate [86]. RNA was mixed with 1.5 μl RPPH enzyme and 2 μl 10X NEB Buffer 2 in a final volume of 20 μl. This was incubated at 37˚C for 1 hour. The number of PCR cycles was increased

from 11 to 15 as per protocol optimised previously [35] otherwise all steps were done according to manufacturer's instructions. Size selection was performed using gel extraction with a 6% TBE gel (Invitrogen). Validation of 22G-RNA libraries was done on a 2200 TapeStation (Agilent).

## Library construction

**RNA library construction.** cDNA library preparation and PolyA-selected RNA sequencing was carried out in the MRC LMS Genomics Facility to generate PE100bp reads. A cDNA library from 31 samples was prepared using the TruSeq RNA Sample Prep kit (Illumina) following the manufacturer's instructions. The cDNA library was then hybridised to the flowcell of the Illumina HiSeq 2500 and sequenced. The library was run on 3 separate lanes in order to achieve good coverage depth. The sequencing data were processed by the instrument's Real Time Analysis (RTA) software application, version 1.18.64. De-multiplexing was done within the sequencing facility with CASAVA.

**ATAC library construction.** For submission to LMS Genomics Facility, multiplexing of samples was performed, with 10–12 samples grouped into a pool. Samples were combined as 10 nM dilutions. Paired end sequencing of ATAC samples was done in the LMS Genomics Facility on a HiSeq instrument. The sequencing data were processed by the instrument's Real Time Analysis (RTA) software application, version 1.18.66.3 using default filter and quality settings. Demultiplexing was done within the sequencing facility using CASAVA- 2.17, allowing zero mismatches.

**Small RNA library construction.** For submission to LMS Genomics Facility, 3 nM dilutions were prepared from each small RNA sample. 3 nM sample dilutions were pooled at ~ 30 samples per pool with unique indexes. Sample pools were validated on TapeStation and quantified on Qubit using dsDNA reagents. MiSEQ Single Read sequencing (50 bp read length) was used to validate sample pools. Balancing or preparation of new pools was undertaken to improve the balance of samples in the pool as needed. High Output single read sequencing was performed on NExtSeq 2000 instrument in the LMS Genomics Facility. The max read length was 60 bpm. The sequencing data were processed by the instrument's Real Time Analysis (RTA) software application, version 2.11.3.

## Pre-processing and alignments

**RNA-seq libraries.** RNA fastq files were aligned to the *C. elegans* genome using Bowtie2 [87] to produce a file containing the raw counts data with the following command;

```
bowtie2 \
-1 ${file1} \
-2 ${file2} \
-x (path to genome)/cel_genomeWS252 \
-S ${Prefix1}.sam
-p 16
```

Sam files were sorted and converted to bam and then bed files. Sorted counts were intersected with coordinates of coding genes derived from the UCSC genome browser [88] using the following BEDTools commands [89].

```
intersectBed -c -a C.elegans_gene_names2.bed -b ${Prefix1}_2.bed \
-wa > ${Prefix1}_intersected_with_genes.bed
```

The files were simplified so that only the longest transcript was represented and 21U-RNAs were removed. In addition, all genes for which no expression was seen in any of the samples were removed. The read values were normalised using DESeq2 [90].

**ATAC-seq libraries.**   Trimming of reads to remove adapters was performed using FASTX Toolkit from the Hannon Lab (RRID:SCR_005534). Sequence alignment was carried out using Bowtie2. [87] to produce a file containing the raw counts data with the following command;

```
bowtie2 \
-1 ${file1}.cut.paired.fq
-2 ${file2}.cut.paired.fq \
-x (path to genome)/cel_genomeWS252
-S ${file1%.fastq}.sam \
-p 8
```

Sam files were sorted and converted to bam and then bed files. Sorted counts were intersected with coordinates for regulatory elements obtained from data produced by the Ahringer Group [50]. Using the following BEDTools commands [89]:

```
bedtools intersect -c -a Promoter_enhancer.gff \
-b ${Prefix1}_2.sorted.bed > ${Prefix1}_promoter_enhancer.bed
```

ATAC-seq across regulatory element loci was then annotated with *C. elegans* gene names obtained from the UCSC genome browser [88]. This resulted in quantification of the number of ATAC-seq reads mapping to each regulatory element. These counts were normalized using the DESeq2 pipeline [90].

**Small RNA.**   Secondary processing and demultiplexing of data was done within the sequencing facility with Bcl2fastq 2_2.20.0 (Illumina). Adapters were removed using FASTX Toolkit from the Hannon Lab (RRID:SCR_005534). For 22G-RNA mapping the un-collapsed reads were aligned to the Ce11 genome using Bowtie [91] allowing 0 mismatches. 22G-RNAs were selected using a perl script and antisense reads mapping to transcribed regions corresponding to protein-coding genes were selected using Bedtools [89]. Reads mapping to each transcribed gene were summed. Data was normalized using DESeq2. For piRNAs and miRNAs, reads were collapsed using FASTX Toolkit from the Hannon Lab (RRID:SCR_005534) and exact matching to annotated piRNAs [63] and miRNAs [92] was performed using the seqinr package. Simulations to find the expected number of epimutations with a cutoff of a Z-score of 2.25 were performed by randomly populating a table with epimutations at the observed average rate and counting the number that were inherited for 1 generation.

## Identification of epimutations

Following normalisation of count values, a linear model was created in which $\log_2$(fold changes) were plotted against the $\log_2$(mean counts) for each gene in line with the approach taken in previous work to identify epimutations in small RNAs [35]. For each data point, a Z-score was derived by subtracting the mean of the residuals from the linear model from each individual residual value and dividing the output by the standard deviation of the residuals. Regions with a Z-score greater or less than 2.25 were defined as showing differential accessibility. Regions with a Z-score greater than 2.25 were annotated as 'Up' epimutations and regions with a Z-score less than 2.25 were annotated as 'Down' epimutations.

## Test for epimutation inheritance

Simulation of data with random distribution of epimutations across a range of Z-scores was performed as described above. Briefly, we applied each value from a range of Z-score cut offs (1–3) to the ATAC-seq Z-score data set to identify epimutations and produce a corresponding set of binarized data tables in which either 1 (Up epimutation) or -1 (Down epimutation) indicated an epimutated locus at a specific generational time point. As would be expected, sequentially raising the threshold for data points to qualify as epimutations imposed increased

stringency thereby reducing the number of points which qualified as epimutations, while the lowest and therefore most permissive threshold qualified a larger number of points as epimutations. For each binarized data table, we then produced a large number of simulated data tables for which the rate of epimutations arising per generation in the real data set was preserved, but the genomic sites undergoing epimutation was determined through random re-assortment of epimutations across loci. Random re-assortment was restricted to loci deemed to be 'epimutable', i.e. those with epimutations in the real data set. This was to prevent the confounding effect of incorporating loci which are never epimutated. Therefore, we examined whether epimutations arose in consecutive generational time points in the same locus, as would be expected if a mechanism for inheritance is active. We were able to reject the null hypothesis that epimutations were no more likely to arise in consecutive generations in real data compared to simulated data.

## Removal of any epimutations which potentially could have been the result of environmental factors

We recognised that despite tightly controlled conditions, minor changes in the experiment environment could occur as the mutation accumulation experiments ran over an approximate twelve-week period. As the experiment was conducted in a standard laboratory setting, slight changes to UV light levels and ambient temperature over the course of the experiment could potentially affect worms when outside of the incubator. These could constitute external stimuli which could potentially have induced epimutations. Therefore, these would not represent spontaneous changes. We reasoned that such effects would feature in all generations similarly as the mutation accumulation lines were run in parallel and handled outside of the incubator at the same time points. In order to remove epimutations which may have been induced, we excluded any epimutations which had the same onset time and duration in all three lineages.

## Quantification of epimutation duration

Our data derived from a pre-mutation founder generation and alternate generation time points (even numbers; generations 2, 4, 6, 8, 10, 12, 14, 16, 18 and 20) from a 20 generation lineage. This required us to form certain assumptions regarding the epimutation status of the intervening generations for which no data was collected. In addition, certain generations were missing from several of the data sets. RNA-seq and small RNA-seq data sets for MA line B lacked generation 4, and for MA line C lacked generation 14. ATAC-seq data set for MA line B lacked generation 4, and for MA line C lacked generation 8. Both ATAC-seq and RNA-seq data sets were complete for MA line A. The reason for missing generations was insufficient yield of RNA or cDNA from these generations, therefore these samples were omitted from sequencing.

When examining the length of epimutations, we assumed that intervening odd numbered generations (1, 3, 5, 7, 9, 11, 13, 15, 17, 19) were not epimutated unless the even numbered generations either side were epimutated. In order to be considered to be inherited, we required that two consecutive even numbered generations displayed an epimutation. This is therefore a conservative approach to detecting inherited epimutations. In measuring the duration of epimutations we accounted for the intervening odd number generations which were assumed to be epimutated. For example, for the germline coding promoter which maps to gene Y65B4BL.7 (coordinates: chr1:510778:510928), the epimutation status over the course of Lineage A was as follows (Table 4):

This locus is recorded as having 3 separate epimutations. 1 putative epimutation (down) which arose in Generation 2 and had a length of 1, i.e. non-inherited. 1 epimutation (down)

**Table 4. Y65B4BL.7 Chromatin-based epimutation status over 20 generations in Lineage A.**

| Generation | 2 | 4 | 6 | 8 | 10 | 12 | 14 | 16 | 18 | 20 |
|---|---|---|---|---|---|---|---|---|---|---|
| Epimutation status | -1 | 0 | 0 | 0 | -1 | -1 | -1 | 0 | -1 | 0 |

which arose in Generation 10 and had a length of 5 (1 + 4 additional generations until disappeared in Generation 14), and finally a non-inherited putative epimutation (down) which arose in Generation 18 and had a length of 1. Therefore, for the epimutation running across Generations 10, 12 and 14, we assumed that epimutations are present in the intervening odd numbered generations 11 and 13.

To be considered as within a single epimutation run, we required that epimutation data points have the same directionality, i.e. all -1 or all 1. A transition from -1 to 1 or vice versa was considered to be a termination of the preceding epimutation and commencement of a new epimutation run. We reasoned that epimutation inheritance would also include inheritance of direction in which chromatin state changed as molecular processes establishing heterochromatin are antagonistic to those dismantling heterochromatin.

We applied a linear model using the lme4 package in RStudio [93] to obtain estimates for the effect of each gene on epimutation duration. We used K-means clustering of the estimates to identify the subset of genes predicted to cause 'long-lived' changes in RNA-seq, ATAC-seq and small RNA data sets. The remaining loci in each data set were separated into those with short-lived inherited changes and those with non-inherited changes. The three 'test' lists of genes (long-lived, short-lived and non-inherited) were obtained for RNA-seq, ATAC-seq and small RNA data sets and compared to genes without epimutations in subsequent gene set enrichment analyses (below).

## Integration of gene expression change and chromatin epimutation data

In order to match up gene expression changes with chromatin states at associated regulatory elements we were required to deal with the unequal ratio of genes to regulatory elements. As would be expected, our data comprised a larger number of regulatory elements than genes. Certain genes have multiple associated coding promoters and putative enhancers as defined previously [50,94–96]. The *C. elegans* lifespan regulating gene *sesn-1* [97], for example, has 3 annotated coding promoters and 12 annotated putative enhancers.

We integrated ATAC-seq data with RNA-seq data in a gene-centric manner. For each gene we recorded its expression state and if expression state change was inherited and the presence of any epimutated regulatory elements. Inheritance of epimutation was recorded if an epimutation was present in at least two consecutive generations for any single regulatory element. We also looked for evidence of simultaneous epimutations in associated regulatory elements. For example, for *C. elegans* detoxification gene *ugt-26* [69] the expression status (Table 5) and associated regulatory element epimutation status (Table 6) over the course of Lineage A were as follows:

**Table 5. Gene expression status for *ugt-26*.**

| Generation | 2 | 4 | 6 | 8 | 10 | 12 | 14 | 16 | 18 | 20 |
|---|---|---|---|---|---|---|---|---|---|---|
| Epimutation status | 0 | 1 | 0 | 1 | 1 | 1 | 1 | 1 | 1 | 1 |

**Table 6. Chromatin-based epimutation status for *ugt-26*.**

| Generation | 2 | 4 | 6 | 8 | 10 | 12 | 14 | 16 | 18 | 20 |
|---|---|---|---|---|---|---|---|---|---|---|
| Epimutation status | 0 | 0 | 0 | 1 | 1 | 0 | 0 | 0 | 1 | 0 |

Therefore, *ugt-26* was recorded as having both inherited expression changes (onset Generation 8 running to at least Generation 20) which were synchronised with chromatin state changes in an associated regulatory element, which are themselves inherited (onset Generation 8, offset Generation 10). Importantly, we did not require that simultaneous epimutations persist for the same duration as inherited expression changes. This is so as to take into account the potential that once established, epigenetic changes may persist in the absence of the initiating mechanism [41].

## Gene set enrichment analyses

Lists of epimutated or expression changed genes (test lists) were uploaded to WormEnrichr [98,99] and the number of genes annotated with each identified ontology term was compared between the test lists and background sets of genes which had no epimutations or expression changes. The background gene set was too large to be uploaded in entirety to WormEnrichr, therefore samples were taken from the list (50 samples each of 6000 genes), analysed for ontology enrichments and compared to the test list. We required a minimum of 5 genes to be present in either the test or background sample gene list for the ontology term to be included in the analysis. Enrichment of test list genes for ontology terms compared to genes sampled from the background list was calculated using a Chi Squared Test. When background lists were small enough to be uploaded in entirety to WormEnrichr, enrichment was calculated with Fisher's Exact Test.

## Production of plots

Plots were made in R Studio (Rstudio Team (2020). Rstudio: Integrated Development for R. Rstudio, PBC, Boston, MA.) with aesthetic editing in Adobe Illustrator (Adobe Illustrator software). Schematic diagrams were made in BioRender (BioRender.com) and Adobe Illustrator.

## Supporting information

**S1 Fig. 22G-RNA class of small RNAs have greatest proportion of heritable epimutations compared to miRNAs and piRNAs, and this is greater than what would be expected by chance.** For each small RNA class (22G-RNAs, mi-RNAs and piRNAS), comparison of proportion of changes exceeding Z-score threshold (2.25, - 2.25) that are inherited to at least two subsequent generations between observed data (dot) and observations derived from 10000 random simulations (Methods).
(TIF)

**S2 Fig. Epimutation rates are similar across three independent worm lineages. A & B.** Gene expression changes. Percentage of genes showing significantly increased **(A)** and decreased **(B)** expression per generation. **C & D.** Chromatin state changes. Percentage of regulatory elements showing significantly increased **(C)** and decreased **(D)** chromatin accessibility per generation. **E & F.** 22G-RNA level changes. Percentage of genes in each lineage showing significantly increased **(E)** and decreased **(F)** antisense 22G-RNA levels per generation. For all plots, box shows the interquartile range with horizontal line at the median; the whiskers extend to the furthest point no more than 1.5 times the interquartile range. Kruskal-Wallis rank sum test gives non-significant p-values > 0.1 for comparison of worm lineages A, B and C within the categories of data type (Gene expression, Chromatin state, 22G-RNA level) and epimutation direction (Up transitions or Down transitions).
(TIF)

**S3 Fig. Deriving subsets of genes with long-lived expression changes and epimutations.** K-means clustering was done to identify the subset of genes predicted from a linear model to have significantly long-lived expression changes **(A)**, chromatin-based epimutations **(B)**, and 22G-RNA-based epimutations **(C)**. The long-lived gene sets were derived independently using an alternative method based on expectation maximisation (EM) clustering using MixTools package in R ('MixEM method'). The overlap between long-lived gene sets derived through both methods is shown for genes with expression changes **(D)**, loci with chromatin-based epimutations **(E)**, and loci with 22G-RNA-based epimutations **(F)**.
(TIF)

**S4 Fig. Small RNA pathways in chromatin and 22G-RNA-based epimutations.** From left to right. Percentage of genes with expression changes (red bar), percentage of regulatory loci with chromatin epimutations (green bar) and percentage of genes with antisense 22G-RNA epimutations (gold bar) which overlap with piRNA cluster genes.
(TIF)

**S5 Fig. Enrichment of direction stratified gene expression changes and epimutations in different constitutive chromatin domains. A.** Bubble plot showing distribution of UP and DOWN non-inherited short-lived and long-lived gene expression changes in distinct chromatin domains. **B.** Bubble plot showing distribution of UP and DOWN non-inherited short-lived and long-lived chromatin-based epimutations in distinct chromatin domains. **C.** Bubble plot showing distribution of UP and DOWN non-inherited short-lived and long-lived 22G-RNA-based epimutations in distinct chromatin domains. For all plots, Y-axis shows constitutive chromatin domains investigated. X-axis shows $\log_2$(Odds) of enrichment. Odds ratios and p-values calculated with Fisher's Exact Test with Bonferroni correction. p-value cut off for significance is 0.1.
(TIF)

**S6 Fig. Differential enrichment of 22G-RNA-based epimutations in small-RNA pathways.** Bubble plot showing enrichment of different small RNA pathways for 22G-RNA-based epimutations of different durations. Y-axis shows specific small-RNA pathway associated proteins. X-axis shows $\log_2$(Odds) of enrichment. Odds ratios and p-values are calculated using Fisher's Exact Test with Bonferroni Correction. p-value cut off for significance is 0.1.
(PDF)

**S7 Fig. Genes with inherited expression changes and simultaneous chromatin-based epimutations are enriched for functions involved in defence response to bacteria.** Y-axis shows ontology terms. X-axis shows $\log_{10}$(Odds) of enrichment. Top 10 results shown. Odds ratios and p-values are calculated using Fisher's Exact Test with Bonferroni Correction. p-value cut off for significance is 0.1.
(TIF)

**S8 Fig. Chromatin fluctuations at X chromosome are not altered by adjusting normalisation thresholds.** *pgp-5*, *pgp-6*, *pgp-7*, *pgp-8* expression and regulatory element chromatin state over 20 generations. Normalisation of ATAC-seq counts is restricted to X chromosome. Data are for worm lineage A. Generational time points on X-axis. Z-score for epimutation status shown on Y-axis with 0 equivalent to PMA state. Horizontal thresholds indicate Z-score cut-offs; > 2.25 = UP epimutation and < - 2.25 = DOWN epimutation.
(TIF)

**S9 Fig. Xenobiotic defence genes fluctuate under relaxed conditions while housekeeping genes do not. A.** Expression over 20 generations across 3 independent worm lineages in

housekeeping genes was compared to that of families of xenobiotic defence genes identified to contain long-lived epimutations. Housekeeping genes n = 13, *pgp* genes n = 14, *nhr* genes n = 268, *ugt* genes n = 66, *cyp* genes n = 74. **B.** As in **(A)** but the 4 gene *pgp* cluster (*pgp-5*, *pgp-6*, *pgp-7*, *pgp-8*) is removed showing that removing this gene cluster removes bias of *pgp* gene family towards Up epimutations as seen in **(A)**.
(TIF)

**S1 Table. Worm Lineage A gene expression, table of Z scores.** File name: S1_Table.csv.
(CSV)

**S2 Table. Worm Lineage B gene expression, table of Z scores.** File name: S2_Table.csv.
(CSV)

**S3 Table. Worm Lineage C gene expression, table of Z scores.** File name: S3_Table.csv.
(CSV)

**S4 Table. Worm Lineage A 22G-RNA levels, table of Z scores.** File name: S4_Table.csv.
(CSV)

**S5 Table. Worm Lineage B 22G-RNA levels, table of Z scores.** File name: S5_Table.csv.
(CSV)

**S6 Table. Worm Lineage C 22G-RNA levels, table of Z scores.** File name: S6 Table.csv.
(CSV)

**S7 Table. Worm Lineage A ATAC counts, table of Z scores.** File name: S7 Table.csv.
(CSV)

**S8 Table. Worm Lineage B ATAC counts, table of Z scores.** File name: S8 Table.csv.
(CSV)

**S9 Table. Worm Lineage C ATAC counts, table of Z scores.** File name: S9_Table.csv.
(CSV)

**S10 Table. Worm Lineage A gene expression Table of Up (1) and Down (-1) epimutations.** File name: S10_Table.csv.
(CSV)

**S11 Table. Worm Lineage B gene expression Table of Up (1) and Down (-1) epimutations.** File name: S11_Table.csv.
(CSV)

**S12 Table. Worm Lineage C gene expression Table of Up (1) and Down (-1) epimutations.** File name: S12_Table.csv.
(CSV)

**S13 Table. Worm Lineage A ATAC Table of Up (1) and Down (-1) epimutations.** File name: S13_Table.csv.
(CSV)

**S14 Table. Worm Lineage B ATAC Table of Up (1) and Down (-1) epimutations.** File name: S14_Table.csv.
(CSV)

**S15 Table. Worm Lineage C ATAC Table of Up (1) and Down (-1) epimutations.** File name: S15_Table.csv.
(CSV)

**S16 Table. Worm Lineage A 22G-RNA Table of Up (1) and Down (-1) epimutations.** File name: S16_Table.csv.
(CSV)

**S17 Table. Worm Lineage B 22G-RNA Table of Up (1) and Down (-1) epimutations.** File name: S17_Table.csv.
(CSV)

**S18 Table. Worm Lineage C 22G-RNA Table of Up (1) and Down (-1) epimutations.** File name: S18_Table.csv.
(CSV)

**S19 Table. Table of values used in Fig 1C.** File name: S19_Table.csv.
(CSV)

**S20 Table. Table of values used in Fig 2A.** File name: S20_Table.csv.
(CSV)

**S21 Table. Table of values used in Fig 2B.** File name: S21_Table.csv.
(CSV)

**S22 Table. Table of values used in Fig 3B.** File name: S22_Table.csv.
(CSV)

**S23 Table. Table of values used in Fig 3C.** File name: S23_Table.csv.
(CSV)

**S24 Table. Table of values used in Fig 3D.** File name: S24_Table.csv.
(CSV)

**S25 Table. Table of values used in Fig 3E.** File name: S25_Table.csv.
(CSV)

**S26 Table. Table of values used in Fig 4A.** File name: S26_Table.csv.
(CSV)

**S27 Table. Table of values used in Fig 4B.** File name: S27_Table.csv.
(CSV)

**S28 Table. Table of values used in Fig 4C.** File name: S28_Table.csv.
(CSV)

**S29 Table. Table of values used in Fig 5A.** File name: S29_Table.csv.
(CSV)

**S30 Table. Table of values used in Fig 5B.** File name: S30_Table.csv.
(CSV)

**S31 Table. Table of values used in Fig 5C.** File name: S31_Table.csv.
(CSV)

**S32 Table. Table of values for gene expression used in Fig 6A and 6B.** File name: S32_Table.csv.
(CSV)

**S33 Table. Table of values for chromatin state used in Fig 6A.** File name: S33_Table.csv.
(CSV)

**S34 Table. Table of values for 22G-RNA level used in Fig 6B.** File name: S34_Table.csv.
(CSV)

**S35 Table. Table of values used in S2A Fig** File name: S35_Table.csv.
(CSV)

**S36 Table. Table of values used in S2B Fig** File name: S36_Table.csv.
(CSV)

**S37 Table. Table of values used in S2C Fig** File name: S37_Table.csv.
(CSV)

**S38 Table. Table of values used in S2D Fig** File name: S38_Table.csv.
(CSV)

**S39 Table. Table of values used in S2E Fig** File name: S39_Table.csv.
(CSV)

**S40 Table. Table of values used in S2F Fig** File name: S40_Table.csv.
(CSV)

**S41 Table. Table of values used in S3A Fig** File name: S41_Table.csv.
(CSV)

**S42 Table. Table of values used in S3B Fig** File name: S42_Table.csv.
(CSV)

**S43 Table. Table of values used in S3C Fig** File name: S43_Table.csv.
(CSV)

**S44 Table. Table of values used in S3D Fig** File name: S44_Table.csv.
(CSV)

**S45 Table. Table of values used in S3E Fig** File name: S45_Table.csv.
(CSV)

**S46 Table. Table of values used in S3F Fig** File name: S46_Table.csv.
(CSV)

**S47 Table. Table of values used in S4 Fig** File name: S47_Table.csv.
(CSV)

**S48 Table. Table of values used in S5A Fig** File name: S48_Table.csv.
(CSV)

**S49 Table. Table of values used in S5B Fig** File name: S49_Table.csv.
(CSV)

**S50 Table. Table of values used in S5C Fig** File name: S50_Table.csv.
(CSV)

**S51 Table. Table of values used in S6 Fig** File name: S51_Table.csv.
(CSV)

**S52 Table. Table of values used in S7 Fig** File name: S52_Table.csv.
(CSV)

**S53 Table. Table of values for gene expression used in S8 Fig File name: S53_Table.csv.**
(CSV)

**S54 Table. Table of values for chromatin state used in S8 Fig File name: S54_Table.csv.**
(CSV)

**S55 Table. Table of values used in S9 Fig File name: S55_Table.csv.**
(CSV)

**S56 Table. Table of values used in Table 1.** File name: S56_Table.csv.
(CSV)

**S57 Table. Table of values used in Table 2.** File name: S57_Table.csv.
(CSV)

## Acknowledgments

We thank Prof. Vahid Shahrezaei (Department of Biomathematics, Imperial College London) for guidance on the statistical approach used to identify epimutations, Dr. Subhanita Ghosh for assistance in developing the ATAC-seq protocol and members of the Epigenetics and Evolution group for helpful comments on the manuscript.

## Author Contributions

**Conceptualization:** Rachel Wilson, Peter Sarkies.

**Formal analysis:** Rachel Wilson, Peter Sarkies.

**Investigation:** Rachel Wilson, Maxime Le Bourgeois, Marcos Perez, Peter Sarkies.

**Methodology:** Rachel Wilson, Peter Sarkies.

**Supervision:** Peter Sarkies.

**Writing – original draft:** Rachel Wilson, Peter Sarkies.

**Writing – review & editing:** Rachel Wilson, Marcos Perez, Peter Sarkies.

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
