## [Decision Letter · Decision Letter 0]

30 Nov 2022

Dear Dr Sarkies,

Thank you very much for submitting your Research Article entitled 'Fluctuations in chromatin state at regulatory loci occur spontaneously under relaxed selection and are associated with epigenetically inherited variation in C. elegans gene expression.' to PLOS Genetics.

The manuscript was fully evaluated at the editorial level and by independent peer reviewers. The reviewers appreciated the attention to an important topic but identified some concerns that we ask you address in a revised manuscript.

We therefore ask you to modify the manuscript according to the review recommendations. Your revisions should address the specific points made by each reviewer.

Yours sincerely,

Marnie E. Blewitt

Academic Editor

PLOS Genetics

Wendy Bickmore

Section Editor

PLOS Genetics

Reviewer's Responses to Questions

**Comments to the Authors:**

Reviewer #1: This is a follow-up to the paper published in Nat Ecol Evol authored by the same group in 2020. In that paper, they showed that epimutations driven by small RNAs arise spontaneously, are 25x more common than mutations, and most persist for up to 3 generations. In the present manuscript, they show that heritable epimutations driven by chromatin changes also occur and that most persist for only a few generations. They are also rarer than mutations and rarer than epimutations driven by small RNAs. These epimutations are reflected in gene expression changes, possibly involving a xenobiotic response.

This work is important because the evolutionary significance of transgenerational inheritance is unclear. Most of the past research has focused on models of transgenerational inheritance that are quite artificial (e.g., silencing of transgenes). In this work, the authors use C. elegans mutation accumulation (MA) lines, which evolve neutrally. Since drift dominates over selection in those lines, the stability of epimutations can be accurately measured. Importantly, the phenotypes being assessed are not artificial but are assessed directly from sequencing data. Particularly interesting is the fact that the phenotypic variation assessed has the potential to be of adaptive value. Namely, variation in response to xenobiotics may provide a fast response to changes in the environment.

The manuscript is clearly written and the conclusions were based on data provided in the paper. I do not have any concerns in terms of interpretations of the data or statistical analysis.

Minor:

Line 630: Update the paper to Johnson, L.M., Smith, O.J., Hahn, D.A., and Baer, C.F. (2020). Short-term heritable variation overwhelms 200 generations of mutational variance for metabolic traits in Caenorhabditis elegans. Evolution 74, 2451-2464. https://doi.org/10.1111/evo.14104.

Figure 5: the two greens are too similar. Use colours that are easier to distinguish.

Reviewer #2: The manuscript by Wilson et al describes a study following epigenetic changes in gene expression, chromatin accessibility and 22G-RNA populations in three independent C. elegans lineages propagated at minimum population size (2) for 20 generations.

This is the first study analyzing chromatin epimutations, to my knowledge. Indeed, the authors found heritable changes in chromatin compaction that mostly lasted for three generations but some lasted over 10 generations. Moreover, heritable changes in gene expression were significantly more frequent at genes exhibiting chromatin epimutation. The authors defined separate groups of genes with short-lived and long-lived epigenetic changes in gene expression, 22G-RNA levels, and chromatin accessibility. This allowed them to look at the enrichment of functional gene categories among these groups and highlight the enrichment of adaptive response pathways. Finally, four gene families with critical roles in xenobiotic defense that showed long-lived expression changes and long-lived chromatin changes (but not long-lived 22G-RNA changes) were identified. This strongly suggests that chromatin epimutations play a role in adaptation to environment.

Apart from the main story line described above, the authors generated valuable resource for future analyses and experimental studies. Their finding that heritable gene expression changes are associated with both concordant and discordant changes between gene expression and chromatin epimutations is intriguing, so is the finding that specific chromatin environments are associated with distinct heritable epimutations (gene expression, chromatin, 22G-RNA levels).

Overall, the manuscript is well written and will be of great interest to the readers of PLoS Genetics.

Points for improvement:

1) In Figure 4 it will be informative to see the directionality of changes: whether “UP” or “DOWN” changes drive the enrichment or depletion.

2) Figure 1C legend or corresponding text. It should be spelled out that “UP” counts of ATAC-seq correspond to increased opening of chromatin and “DOWN” counts reflect chromatin closing.

3) There should be more precision in describing what is subject to epimutation (22G-RNA abundance, gene expression, chromatin accessibility) without implying functional consequences. This is most relevant to 22G-RNAs. Some examples are below.

Line 12: “Changes in epigenetic regulation termed epimutations”. I suggest: ““Changes in epigenetic regulators…”

Line 14: “Epimutations mediated by small non-coding RNAs”. What are those epimutations: gene expression changes? Phenotypes? Corrections: “Epimutations in 22G-RNA pools” or “epimutations carried by small non-coding RNAs”.

Also on line 67: “epimutations mediated by 22G-RNAs arise spontaneously”. Rather “epimutations in 22G-RNA levels”, “epimutations driving 22G-RNA abundance arise spontaneously”.

Check paragraph starting on line 331.

Line 505: “… 22G-RNAs drive spontaneous short-lived epimutations” is not correct based on the cited publication. Instead: “22G-RNA levels are subject to spontaneous short-lived epimutations”.

Lines 134, 136, 221, Fig 2 legend: “chromatin mediated epimutations”. Chromatin mediates epimutations of what? Correct: “epimutations in chromatin accessibility” or “epimutations carried via chromatin organization”.

Line 17: “mechanism for transgenerational epigenetic inheritance.” Inheritance of what? Apparently, gene expression changes, based on the manuscript. Traditionally, epigenetic inheritance of phenotypes is implied.

Line 226: “epimutations in piRNA loci” change to “epimutations in piRNA expression levels”

4) Discussion relevant to Figure 7 can be more organized. Line 599-600 and Figure 7: “mechanism of epimutation inheritance itself may become more robust under stress” is vague, suggest how this may happen. For example, Scenario 1 is consistent with induced epimutations causing specific changes in gene expression that become more frequent in the population and more heritable. Scenario 2 reflects selection of existing advantageous epimutations.

5) I suggest that the authors provide data tables in a more accessible format, such as Excel.

Reviewer #3: This work explores the functional relevance of spontaneous epimutations, or changes in epigenetic regulation independent of changes to DNA sequence. Previous work has shown that populations of wildtype worms can accumulate epimutations mediated by spontaneous changes to 22G-RNA profiles (Beltran et al., 2020), and that these epimutations can persist for several generations. However, previous work was unable to identify the functional relevance of a high epimutation rate, and thus this study seeks to explore other mechanisms of epigenetic regulation that may contribute to their evolutionary significance. In particular, the authors explore changes to chromatin architecture via ATAC-seq, as well as changes to both mRNA and small RNA profiles, over 20 generations of a Mutation Accumulation (MA) experiment for 3 independently maintained worm lineages. They observe spontaneous changes to chromatin state at many loci (~1% of regulatory regions), as well as many changes in gene expression. In some cases, these events appear correlated, yet in many, they appear anti-correlated (i.e. some gene expression increases correspond counterintuitively to “closing” chromatin state, and vice versa - I believe this is the conclusion of Table 1 and Figure 3, see criticism below). Notably, genes with spontaneous expression changes that are inherited are more likely to be associated with simultaneous chromatin epimutations than genes with non-inherited expression changes (Fig 3B). This does support a correlation between chromatin state changes and persistent gene expression changes. Further, their work finds that xenobiotic defense responses are enriched among the categories of genes that undergo “long-lived” changes to gene expression (i.e. the expression changes persist for over 10 generations), as well as in the categories of genes that undergo “long lived” changes to chromatin state (Fig 4A,B). This supports a hypothesis that one of the functional, evolutionary advantages of maintaining a high spontaneous epimutation rate is to increase potential fitness in response to pathogenic or other environmental stresses. Overall, the conclusions appear to be supported by the data. However, substantial revisions are required to more clearly and concisely communicate their findings.

Experimental and Major Comments:

1. Figure 1C introduces the criteria and thresholds of the ATAC-seq data by which changes in chromatin are considered a “chromatin epimutation” for subsequent analysis. This figure also introduces UP and DOWN as categories of chromatin state epimutation, but the text does not place these in functional context. What do these designations mean for how chromatin accessibility has changed? Are we to assume that an UP epimutation indicates more accessible chromatin and DOWN epimutation indicates more closed? The functional conclusions of 1C are not thoroughly discussed, even though they are critical for interpreting the subsequent data.

a. Especially regarding Figure 3 and Table 1, the introduction of discordant vs concordant UP and DOWN chromatin changes is extremely confusing, in part due to incomplete explanation. Table 1 as a stepwise analysis of gene expression vs chromatin state is not well explained – the “Background” column is not defined in-text, nor is it clear what the “Background” categories mean relative to the “Association tested” category, nor is it adequately discussed how to interpret these results. The calculations performed to get the reported “Odds ratio” is not well defined. While the basic conclusions drawn in Figure 3B do appear clear and robust, Table 1 requires that we simply “take the author’s word for it” because, as currently presented, it is nearly impossible to interpret.

b. The above criticism holds for Table 2 and Figure 3C, which performs a similar analysis comparing gene expression changes to 22G-RNA epimutations.

c. Figures 3D and 3E both build from the observations in Tables 1 and 2. Now the authors present pairwise analysis between UP and DOWN trends (presumably with reference to Fig 2C, although 2C only discusses chromatin changes and here they also include gene expression changes and 22G-RNA changes), and “concordant” vs “discordant” changes (in reference to whether the compared changes occur in the same direction) to observe the odds of any 2 simultaneous epimutation events being inherited. Contextualizing what UP and DOWN functionally mean for a concordant or discordant change in 1) gene expression, 2) chromatin, and 3) 22G-RNA epimutations is essential to interpreting this figure and to supporting the conclusions of the paper.

2. Lines 387-391 indicate that epimutations at 22G-RNA level (Fig 4C) at the X chromosome are enriched. According to the figure, “non-inherited” and “short lived” are enriched, but “long-lived” are close to zero and not significant. Lines 390-391 state “Both long-lived and short-lived epimutations were similarly enriched” and it is unclear if this still refers specifically to the 22G-RNA level changes at the X chromosome (per the previous sentence), but if so, then it is untrue.

3. The paragraph starting on line 393 discusses enrichment trends in the chromatin state change epimutations (Figure 4B). It states “piRNA clusters… were strongly enriched for epimutations at the level of chromatin.” – it’s unclear if this refers to directly to Figure 4B, but if so it does not appear supported by the data. The only bubble visible for piRNA cluster genes under chromatin state changes (bottom of 4B) is for “non-inherited” and it lies very close to zero, indicating it is not a strongly enriched category for any inherited changes at all. Further,

4. This paragraph also refers to Supplemental Figure 4 which does in fact appear to support their stated conclusion, although this suggests a conclusion contrary to Figure 4B, so acknowledgement of where that discrepancy arises would be important.

5. Figure 5 identifies the cellular pathways enriched for epimutations associated with gene expression changes (5A), chromatin state changes (5B), and 22G-RNA level changes (5C), and sets up for the conclusion that environmental stress and pathogen responses could be an evolutionary reason to allow for high epimutation rates under normal growth conditions. However, these defense pathways are only 1 among many different pathways with equivalent enrichment, yet the others are not particularly discussed. I think the fact that across an array of pathways, “long-lived” epimutations are very highly enriched as compared to “non-inherited” or “short-lived” specifically at the chromatin state level (5B) is extremely interesting and perhaps should be emphasized. This strongly supports how chromatin-based epigenetic changes can set the stage for heritable changes lasting over 10-generations (compare this to 5A, for example, where “short-lived” and “long-lived” changes behave similarly for gene expression changes).

6. Figure 6 identifies a cluster of P-glycoprotein genes that previous studies have already shown to be expressed and behave similarly, and then shows that they behave similarly with expression patterns that change as a unit, as expected. Expanded upon below, Figure 6 is an interesting observation but lends little support to the full article’s conclusions, especially with regards to chromatin state epimutations.

a. The authors have no testable explanation as for why this cluster of genes suddenly increased in expression upon onset of the experiment in all 3 worm lineages (6C), which makes it an interesting observation but I’m unsure how it can be used to support their conclusions.

b. Despite this unexplained robust increase in expression, none of the simultaneously observed chromatin changes at the cluster pass the threshold for UP or DOWN (6C). I recognize that they address this in the text, stating that chromatin activity still “appeared to broadly track with the peaks and troughs of gene expression change” (line488-489), but that association appears weak and overall does not particularly support their conclusions that changes in chromatin state are associated with epimutations in gene expression level.

c. Further, they go on to try and address this discrepancy by adjusting their threshold settings, and this still does not appear to make even a single time point for promoter 1, 2, or 3 chromatin states pass the Z-score significance threshold (supplemental figure 6C).

d. In fact, Lines 484-485 state “We did not observe any epimutations in 22G-RNAs that could explain this”, referring to the co-regulation of this P-gp cluster. Yet, supplemental figure 6B shows a steep drop in 22G-RNAs targeting this cluster that corresponds exactly in time with the increase in cluster gene expression. This dip in 22G-RNAs also does not pass the Z-score threshold (as the chromatin state changes did not either), but appears it may account for the change in cluster expression more robustly than the weak correlation observed with the peaks and troughs in chromatin state.

Minor Comments:

1. The authors should address the RNAi pathways represented in their 22G-RNA epimutation data. Their previous publication (Beltran 2020) states that canonical silencing pathways (including piRNA, HRDE-1, WAGO-1, ERGO-1, and NRDE-3) were enriched for 22G-RNA epimutations but that the canonical licensing pathway CSR-1 is not. If this remains true, it should be briefly discussed here as well as it is important for interpreting the functional relevance of concordant and discordant changes in gene expression vs 22G-RNA epimutations.

2. Figures 4 and 5: the bubbles indicating “non-enriched”, “short-lived”, and “long-lived” heritability are so close in color as to be difficult to tell apart. Additionally, the dots indicating “not significant” should still be visibly distinguishable and color coordinated.

3. Line 452-453 references a conclusion drawn from a “previous observation” – what figure does this refer to?

4. Supplemental Figure 6C is not included in the figure legend.

**Have all data underlying the figures and results presented in the manuscript been provided?**

Reviewer #1: Yes

Reviewer #2: Yes

Reviewer #3: Yes

PLOS authors have the option to publish the peer review history of their article (what does this mean?). If published, this will include your full peer review and any attached files.

Reviewer #1: No

Reviewer #2: No

Reviewer #3: No

---

## [Decision Letter · Decision Letter 1]

1 Feb 2023

Dear Dr Sarkies,

We are pleased to inform you that your manuscript entitled "Fluctuations in chromatin state at regulatory loci occur spontaneously under relaxed selection and are associated with epigenetically inherited variation in C. elegans gene expression." has been editorially accepted for publication in PLOS Genetics. Congratulations!

Yours sincerely,

Marnie E. Blewitt

Academic Editor

PLOS Genetics

Wendy Bickmore

Section Editor

PLOS Genetics

Comments from the reviewers (if applicable):

Reviewer's Responses to Questions

**Comments to the Authors:**

Reviewer #1: I am pleased with the modifications made by the authors.

Reviewer #2: I am satisfied with the clarifications made by the authors, the manuscript reads much better now.

Reviewer #3: The authors' revisions have greatly clarified their message and, overall, the manuscript is acceptable for publication in PLoS Genetics. However, I do have several suggestions for additional edits that might further strengthen their study. These suggestions are optional.

relevant text (lines 340-344)

I might suggest using an alternate word in place “license” in this description, as it has a more commonly used definition in the small RNA field describing how the activating argonaute CSR-1 “licenses” genes for expression in the germline.

relevant text (lines 390- 396)

Consider additionally changing the range of the X axis in Figure 5 as you’ve now done in Figure 4. It does improve resolution and highlights the significant changes you discuss.

relevant section of text (lines 528- 545)

The reframing of this section is well written, I have only minor comments below:

a.) Consider re-adding the schematic of the pgp gene cluster to this figure as it had been previously. I think it highlights for less familiar readers how these genes might be regulated as a unit.

b.) Lines 537-539 state “Pgp genes 5, 6, 7 and 8 had long-lived epigenetic changes in all three lineages. The expression patterns for the 4 pgp genes were strongly similar within each lineage (Fig. 6A & B).” However, with the revised figure 6, no other lineages are shown besides Worm lineage A (as had been in previous Fig 6B).

**Have all data underlying the figures and results presented in the manuscript been provided?**

Reviewer #1: Yes

Reviewer #2: Yes

Reviewer #3: Yes

PLOS authors have the option to publish the peer review history of their article (what does this mean?). If published, this will include your full peer review and any attached files.

Reviewer #1: No

Reviewer #2: No

Reviewer #3: No

**Data Deposition**

http://datadryad.org/submit?journalID=pgenetics&manu=PGENETICS-D-22-01168R1

**Press Queries**

---

## [Editor Report · Acceptance letter]

24 Feb 2023

PGENETICS-D-22-01168R1 

Fluctuations in chromatin state at regulatory loci occur spontaneously under relaxed selection and are associated with epigenetically inherited variation in C. elegans gene expression. 

Dear Dr Sarkies, 

We are pleased to inform you that your manuscript entitled "Fluctuations in chromatin state at regulatory loci occur spontaneously under relaxed selection and are associated with epigenetically inherited variation in C. elegans gene expression." has been formally accepted for publication in PLOS Genetics! Your manuscript is now with our production department and you will be notified of the publication date in due course.

With kind regards,

Zsuzsanna Gémesi

PLOS Genetics

On behalf of:
